



# Interfacial supercooling and the precipitation of hydrohalite in frozen NaCl solutions by X-ray absorption spectroscopy

Thorsten Bartels-Rausch[1], Xiangrui Kong[1*], Fabrizio Orlando[1**], Luca Artiglia[1], Astrid Waldner[1], Thomas Huthwelker[2], Markus Ammann[1]

[1]Laboratory of Environmental Chemistry, Paul Scherrer Institut, Villigen PSI, Switzerland
[2]Swiss Light Source (SLS), Paul Scherrer Institut, Villigen PSI, Switzerland
* now at: Department of Chemistry and Molecular Biology, Atmospheric Science, University of Gothenburg, Gothenburg, Sweden
** now at: Omya International AG, Oftringen, Switzerland

*Correspondence to*: Thorsten Bartels-Rausch (thorsten.bartels-rausch@psi.ch)

**Abstract.** Laboratory experiments are presented on the phase change at the surface of sodium chloride – water mixtures at temperatures between 259 K and 240 K. Chloride is a ubiquitous component of polar coastal surface snow. The chloride embedded in snow is involved in reactions that modify the chemical composition of snow as well as ultimately impact the budget of trace gases and the oxidative capacity of the overlying atmosphere. Multiphase reactions at the snow – air interface have found particular interest in atmospheric science. Undoubtedly, chemical reactions proceed faster in liquids than in solids; but it is currently unclear when such phase changes occur at the interface of snow with air. In the experiments reported here, a high selectivity to the upper few nanometres of the frozen solution – air interface is achieved by using electron yield near-edge X-ray absorption fine structure (NEXAFS) spectroscopy. We find that sodium chloride at the interface of frozen solutions, which mimic sea-salt deposits in snow, remain as supercooled liquid down to 240 K. Below this temperature, hydrohalite exclusively precipitates, anhydrous sodium chloride is not detected. In this work, we present the first NEXAFS spectrum of hydrohalite. The hydrohalite is found to be stable while increasing the temperature towards the eutectic temperature of 253 K. Taken together, this study reveals no differences in the phase changes of sodium chloride at the interface as compared to the bulk. That sodium chloride remains liquid at the interface upon cooling down to 240 K, which spans the most common temperature range in Polar marine environments, has consequences for interfacial chemistry involving chlorine as well as for any other reactant for which the sodium chloride provides a liquid reservoir at the interface of environmental snow. Implications for the role of surface snow on atmospheric chemistry are discussed.





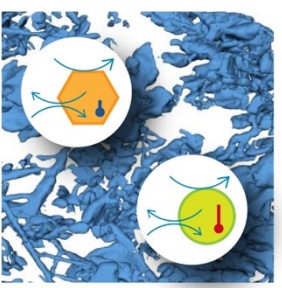


Abstract Teaser

## 1 Introduction

Chemical cycling of halogens affects the composition of the troposphere and via this effect influences climate and impacts
human health (Simpson et al., 2007; Abbatt et al., 2012; Saiz-Lopez and von Glasow, 2012; Simpson et al., 2015). Taken the
abundance of chloride in the form of sea-salt over wide areas of the globe, the atmospheric chemistry of chlorine has long
raised interest in a number of multiphase reactions that liberate chloride into chlorine species in the gas phase. Chlorine has a
direct role as a sink for ozone. Further, reactive chlorine species act as a powerful oxidant on atmospheric cycles that destroy
or produce ozone and are relevant for the atmospheric oxidation capacity (Finlayson-Pitts, 2003; Thornton et al., 2010).
Atmospheric ozone is of concern because it directly impacts atmospheric composition, health, and climate (Simpson et al.,
2007). Prominent example of these reactions in the atmosphere with chloride in sea-salt or salt-dust are shown in Fig. 1. Acid
displacements, where the nitric or sulfuric acid are taken up into the aqueous phase, lead to the emission of hydrochloric acid
to the gas phase. Further examples are reactions with OH radicals (Knipping et al., 2000), and with nitrogen oxides (Osthoff
et al., 2008). Common to all of these reactions is the generation of chlorine gases that either react with OH radicals or photolyze
at wavelengths available in the troposphere to generate reactive chlorine. Given their importance, these multiphase reactions
have intensively been investigated in laboratory studies over the last decade (Abbatt et al., 2012; Saiz-Lopez and von Glasow,
2012; Simpson et al., 2015). One outcome of these investigations is the importance of reactions at the liquid – air interface as
compared to those proceeding in the bulk. An example is the oxidation of chloride by OH radicals. In contrast to the aqueous
phase reactions, it does not require the presence of acid to proceed fast at the air—water interface (Laskin et al., 2006).
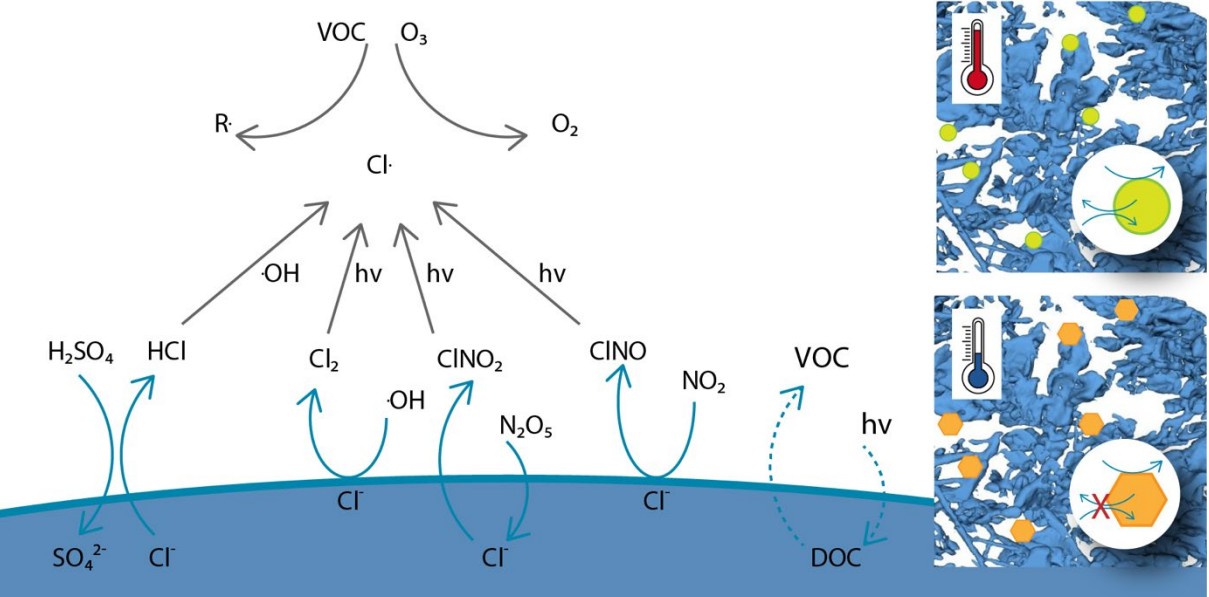

Figure 1: Simplified scheme of multiphase reactions liberating chlorine from sea-salt deposits to the atmosphere and subsequent reactions of chlorine with impacts on the ozone budget and air quality. These reactions may occur both in the bulk phase and at the air-deposit interface for liquid particles. The sea-salt may also maintain a liquid phase for the reaction of other reactants such as the photolysis of organics. The inserts illustrate sea-salt deposits in snow and how phase changes from solid (orange pentagon) to aqueous solutions (green circle) impact the chemical reactivity. See text for details.

The relevance of halogen multiphase chemistry for the atmosphere is not limited to chlorine. A more recent example is the oxidation of bromide. Bromide is present in sea-salt, is a key reactant in ozone depletions in polar atmospheres (Simpson et al., 2015), and participates in atmospheric chlorine chemistry by forming interhalogen compounds (Finlayson-Pitts, 2003). Oldridge and Abbatt (2011) have shown that a Langmuir-Hinshelwood type surface reaction of ozone with bromide occurs at the liquid – air interface simultaneously with a corresponding bulk reaction in the temperature range of 263 K to 248 K. A surface-active reaction intermediate was found to explain the high interfacial reactivity for the case of the reaction with ozone (Artiglia et al., 2017), while other bromine species may directly exhibit surface propensity on their own (Gladich et al., 2020). Clearly, this line of research shows how reaction kinetics and mechanisms differ at the interface from those in the bulk and that heterogeneous chemistry is a key driver in atmospheric chemistry.

In the cryosphere, where the snowpack is strongly impacting the chemistry in the overlaying atmosphere (Dominé and Shepson, 2002; Thomas et al., 2019), halogen compounds are also found within the snow. Sea-salt components, a source of halogens in snow in costal snowpack, might originate from migration from underlying sea-ice or from deposition of wind-transported sea-spray aerosol (Dominé et al., 2004). One characteristic of the cryosphere are its subfreezing temperatures and the consequent



precipitation of chemical constituents at specific temperatures, their eutectic temperature, as also observed in sea-ice (Petrich
and Eicken, 2009). It is known that the precipitations or phase changes of the reactants critically impact the reactivity (Bartels-
Rausch et al., 2014; Kahan et al., 2014; Edebeli et al., 2019). Importantly, it is not only the chlorine chemistry that responds
to the phase of sodium chloride present in frozen systems. Oldridge and Abbatt (2011) showed that the rate of the
heterogeneous reaction of ozone with bromide in sodium chloride -- water mixtures is strongly reduced once sodium chloride
precipitates below 252 K. The author explained this with the reduction in liquid volume that serves as reaction medium for the
bromide in the sample due to the precipitation of sodium chloride. Similar, the photolytical reaction of nitroanisol with pyridine
was found to depend on the amount of liquid in frozen samples and thus to critically respond to precipitation of sodium chloride
(Grannas et al., 2007). Next to its role in atmospheric halogen chemistry, sodium chloride is thus further of importance as salt
to establish and maintain liquid solutions at subfreezing temperature. Its importance arises from its atmospheric abundance but
also because its eutectic temperature of 252 K falls into typical springtime Arctic temperatures – a region and time period
when atmospheric halogen chemistry is most active.

While the phase diagram of sodium chloride – water binary mixtures and the thermodynamic stability domains of salt, solution,
and ice are well known (Koop et al., 2000a), the precise occurrence of nucleation and sodium chloride precipitation is still
debated (Koop et al., 2000a; Wise et al., 2012; Peckhaus et al., 2016). Figure 2 shows a part of phase diagram of sodium
chloride -water mixtures and can be used to illustrate the appearance of the individual phases. Below 251.9 K, the eutectic
temperature of sodium chloride (Koop et al., 2000a), anhydrous solid sodium chloride (NaCl, halite) and solid water (ice) are
the energetically favoured phases. Increasing the temperature of such a hypothetical sample to above 251.9 K, the sodium
chloride will change its phase from solid halite to an aqueous sodium chloride solution. Ice remains the second phase at
temperatures between 251.9 K and 273.2 K. Above 273.2 K, the ice will melt completely and an aqueous sodium chloride
solution will be the only phase present. A characteristic of these systems is that between the eutectic temperature and the
melting point of ice, the system moves along the so-called ice stability line. In other words, the concentration of the sodium
chloride in solution changes with temperature to maintain both phases in equilibrium. At the air-ice interface where both phases
are in equilibrium with water vapour in the gas phase, this can be understood by considering that the aqueous sodium chloride
solution and ice need to have the same vapour pressure. Otherwise, the phase with the higher water vapour pressure would
vanish by evaporation and the one with the lower vapour pressure would grow by condensation. The vapour pressure of ice
decreases with decreasing temperature while that of an aqueous solution decreases with increasing concentration of solute.
With a fixed amount of sodium chloride present in our experiments and also in individual sodium chloride deposits in snow,
consequently the amount of liquid is given by temperature (Koop et al., 2000a). More general, by absorbing and evaporating
water from the surrounding air, composition and volume of the brine will change. Such changes with varying relative humidity
(hygroscopic growth) have long been discussed for aerosol in the troposphere. The changes include phase transitions from the
solid to the liquid solution (deliquescence) and from the liquid to the solid (efflorescence) in aerosol particles at specific
temperature and water partial pressure conditions. Because relative humidity is such a strong driver of the hygroscopic growth,
we base the discussion of phase changes in this work on the relative humidity, a measure for the partial pressure of water, that
the sample was exposed to.

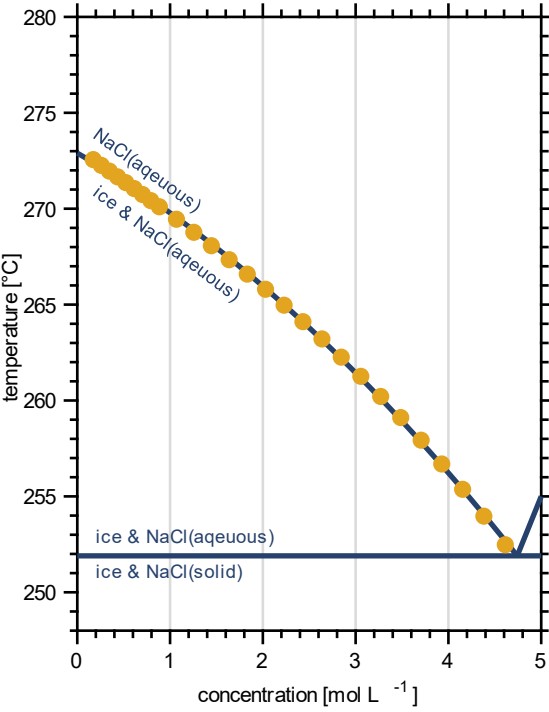


Figure 2: Phase diagram of the NaCl-water binary system. The data show the freezing point depression of sodium-chloride
solutions (yellow filled circles) and give the concentration of an aqueous sodium chloride solution in equilibrium with water
in the temperature range of 273 K to 254 K (Rumble). The dark blue lines indicate the phase boundaries (Koop et al., 2000b;
Rumble), that is it denotes the so-called liquidus and solidus line, respectively, and thus shows the temperature and
concentration range where ice and aqueous sodium chloride solution co-exist. The eutectic temperature of sodium chloride –
water binaries is 251.9 K (Koop et al., 2000a).

Koop et al. (2000a) were the first to show that because precipitation of sodium chloride can be kinetically hindered, i.e.
precipitation may not occur even though temperature has dropped below the eutectic temperature where the solid is the
thermodynamically favoured phase, supercooled sodium chloride solutions in the presence of ice can prevail down to 240 K.
Cho et al. (2002) have observed a liquid fraction in sodium chloride – water mixtures at even lower temperatures of between
228 K and 273 K based on the evaluation of [1]H-NMR signals. Cho et al. (2002) proposed the presence of liquid below the
eutectic temperature to be an interfacial phenomenon, stabilized by surface forces in analogy to the disordered interface
observed for neat ice surfaces when approaching the melting point (Bartels-Rausch et al., 2014).



The goal of this study is to investigate the precipitation and the occurrence of liquid features in sodium chloride – water binary
mixtures in the interfacial region. Both Cho et al. (2002) and Koop et al. (2000a) have applied methods that are not specifically
sensitive to the interface, but are probing the bulk. Near Edge X-ray Absorption Fine Structure (NEXAFS) spectroscopy is
inherently sensitive to the upper few nanometre of interfaces when detecting electrons as done in this work. Oxygen K-edge
NEXAFS spectra of $H_2O$ are an established tool to investigate the hydrogen bonding structure of water and ice with its clear
differences for solid and liquid water (Bluhm et al., 2002; Nilsson et al., 2010; Krepelova et al., 2013; Newberg and Bluhm,
2015; Orlando et al., 2016; Bartels-Rausch et al., 2017; Ammann et al., 2018; Waldner et al., 2018). In our earlier NEXAFS
study (Krepelova et al., 2010a), it was shown by probing the X-ray absorption of oxygen atoms in sodium chloride – water
binary mixtures that the hydrogen bonding network did not reveal the presence of any liquid features at the interface below
the eutectic temperature. Interpretation of these oxygen NEXAFS spectra was complicated by the appearance of crystal water
in the hydrohalite and by the presence of adsorbed $H_2O$. Also, the oxygen spectra might be dominated by ice that is present in
equilibrium with the sodium chloride solution and thus small fractions of liquid might have been difficult to detect. In this
work, we therefore discuss Cl K-edge NEXAFS that have previously been used to inspect the chemical speciation of chlorine
in glasses and in coal (Huggins and Huffman, 1995; Evans et al., 2008). Interest in the local environment of chloride at the
interface in sodium chloride – water binary mixtures comes also from earlier work on nitric and hydrochloric acid adsorbed at
the ice-air interface. We have shown that nitrate and chloride forms solvation shells with a hydrogen-bonding structure similar
to that in aqueous solution in the interfacial region of ice at concentrations low enough to prevent melting (Krepelova et al.,
2010b; Kong et al., 2017). The aim of this study is thus also to investigate the occurrence of solvated chloride at the interface
of sodium chloride – water mixtures at temperatures close to the eutectic. Motivation comes from the role of heterogeneous
chemistry in atmospheric science in general and in particular on the impact that the microchemical environment has on the
reactivity.

## 1 Experimental Part

Experiments were performed at the PHOENIX beam line of the Swiss Light Source (SLS) at the Paul Scherrer Institute using
the Near Ambient Pressure Photoemission (NAPP) set-up previously described (Orlando et al., 2016). NAPP is equipped with
a differentially-pumped electron analyser (Scienta R4000 HiPP-2). The central feature of NAPP is a flow-through cell with a
sample holder the temperature of which is computer-controlled by a flow of cooled helium gas. The measurements were
performed with partial pressures of water between 0.3 mbar and 1.8 mbar in the flow-through cell and temperatures of the
sample between 259 K and 240 K.





## 1.1 Sample Preparation and Water Dosing

To prepare a sample, 1 µl of a 2.12 g sodium chloride (Fluka Trace Select 38979-25G-F) solution in 80 ml of water (Fluka Trace Select 142100-12-F) was dropped at the centre of the sample holder and dried at 60 °C. The sample holder was then moved into the flow-through cell and kept at UHV and at 60 °C to 80 °C for 45 minutes to remove volatile impurities. Water vapour was dosed to the flow-through cell via a 0.8 mm i.d. steel capillary from the vapour above liquid water (Fluka Trace Select 142100-12-F) in a vacuum-sealed, temperature-controlled glass reservoir. Before dosing, the water was degassed by 4 freeze-pump-thaw cycles. Pressure in the flow-through cell was monitored by a capacitance manometer (Baratron 626A) with a measurement range from $5 \times 10^{-4}$ to 10 mbar and an accuracy of 0.25 % of the reading. Temperature was monitored with a Pt-1000 sensor located at the edge of the sample holder. The sensor was calibrated prior to the experiments by growing ice on the sample holder and noting its vapour pressure which is a direct measure of the temperature at the sample spot (Marti and Mauersberger, 1993). During the experiments, the calibration was confirmed when ice was present. At 253 K, the offset between temperature reading and calibration was found to be $4.3 \pm 0.2$ K.

## 1.1 X-ray exited Electron Spectroscopy

Partial Auger-Meitner electron-yield NEXAFS spectra at the Cl K-edge were acquired with a fixed kinetic energy window of 2370 eV to 2390 eV, which includes the KL2,3L2,3 Auger-Meitner peak of chlorine. The pass energy and dwell time were set to 200 eV and 300 ms, respectively. The distance of the sample to the electron analyser inlet (working distance) was 1 mm and the electron analyser was operated with an electron sampling aperture with a diameter of 500 µm. NEXAFS spectra were measured by sweeping the incident X-ray photon energy across the chlorine K-edge from 2815 eV to 2845 eV with steps ranging from 0.2 eV to 1 eV. The NEXAFS spectra were processed by dividing by the photon flux ($I_0$) as derived *in-situ* using a Ni coated membrane, by subtracting the mean pre-edge intensity as background, and by normalising to the mean intensity at 2830 eV to 2833 eV X-ray photon energy. Photoemission spectra (XPS) of O1s, Cl2p, Na1s, C1s, and Au4f were recorded at an incident X-ray photon energy of 2200 eV and a with a pass energy of 100 eV and a dwell time of 120 ms. To quantify, a linear background was applied and the photoemission signal was integrated in Matlab without any peak fitting.

## 1 Results and Discussion

### NEXAFS of brine, halite, and hydrohalite

Figure 3 shows chlorine K-edge X-ray absorption spectra of NaCl salt and of frozen NaCl-water binary mixtures in the presence of ice. NEXAFS spectroscopy probes the X-ray absorption of chlorine atoms, that is the resonant excitation of core electrons into unoccupied molecular orbitals. As exactly those outer orbitals are forming chemical bonds, NEXAFS spectra directly reflect changes to the local chemical environment and structural arrangement. NEXAFS spectroscopy of halogen salts has thus been used to discuss their phase and chemical speciation in geological examples (Huggins and Huffman, 1995; Evans



et al., 2008). In this work, the X-ray absorption spectra were derived by recording the intensity of Auger-Meitner electrons at
2370-2390 eV, which corresponds to the KL2,3L2,3 transition in chlorine (Cleff and Mehlhorn, 1969). Detecting electrons, as
done in this work, makes X-ray absorption inherently surface sensitive, because electrons have a limited escape depth in matter
(Ammann et al., 2018). The escape depth can be quantified by relating it to the inelastic mean free path (IMFP) of electrons in
matter and to the take-off angle of detected relative to the surface normal. The IMFP of electrons with a kinetic energy of
2380 eV is about 7 nm in NaCl and in ice (Tanuma et al., 1991). The take-off angle of electron detection is 30° in our set-up
(Orlando et al., 2016), this gives an escape depth of 6 nm, meaning that cumulatively, 95% of the electrons detected originate
from 18 nm, with an exponentially decreasing contribution from the surface towards the bulk. In the following, we report the
NEXAFS spectra derived from this interfacial region of NaCl-water binary mixtures to discuss changes in the solvation of
chloride by water as we explore the regions of the phase diagram where precipitation of sodium chloride has been described
for bulk samples.

All chlorine K-edge spectra show a strong absorption peak at 2825-2830 eV (region I), the absorption edge, corresponding to
the transition of electrons from the 1s to the empty 4p orbitals.  Halite salt, in Fig. 3 A, shows a strong absorption in region I,
and a characteristic feature at 2838 eV (region II) (Huggins and Huffman, 1995; Evans et al., 2008).  The difference in the
relative intensity of region I as compared to the intensity at photon energies > 2830 eV between Fig. 3A and Huggins and
Huffmann (1995) comes from the latter using fluorescence as detection mode. Detecting X-ray absorption with fluorescence
probes deeper into the bulk of the samples and therefore often suffers from self-absorption which leads to less signal intensities
at the absorption edge (region I).

The small feature at 2821 eV can be assigned to C-Cl bonds that may form in-situ due to reactions involving secondary
electrons and organic impurities (Fujimori et al., 2010). The sample composition in the interfacial region was further
investigated by X-ray photoemission spectroscopy. The C1s and Cl2p spectra support the presence of C-Cl compounds in the
samples (Appendix A): A small feature in the Cl2p spectra at 3 eV higher binding energy than the main chlorine signal might
be attributed to organic carbon species. The C1s photoemission spectra show a feature at 1.5 eV higher binding energy than
the main feature, typical for C-Cl compounds. The main feature of the C1s photoemission spectra is attributed to C-H bonds,
the dominant component of adventitious carbon that forms in a cascade of reactions with secondary electrons during the
experiments. A further source of carbon might be omnipresent traces of carbon in the NaCl and introduced when preparing the
NaCl samples. The atomic ratio of the total carbon to oxygen from water present at the interface was below 0.25, except for
samples in Fig. 3A and F where total carbon to oxygen atomic ratios were 0.5-0.75. Adsorbed water molecules were present
at the interface of all samples, and ice or liquid water formed in some samples, as gaseous water was present in all experiments
with pressures between 0.3 mbar to 1.8 mbar.  The atomic ratio was derived based on the measured C1s/O1s photoemission
intensities and a calibration to account for the analyser efficiency and total X-ray photoionization cross section using C1s/O1s
photoemission intensities of 0.8 mbar $CO_2$ gas following a procedure used before (Krepelova et al., 2013). Direct comparison

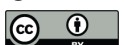

between the individual samples and estimation of surface coverages of the carbon impurities is hampered by the varying water
content at the sample's interface as adsorption and water uptake varies with the individual relative humidity settings.


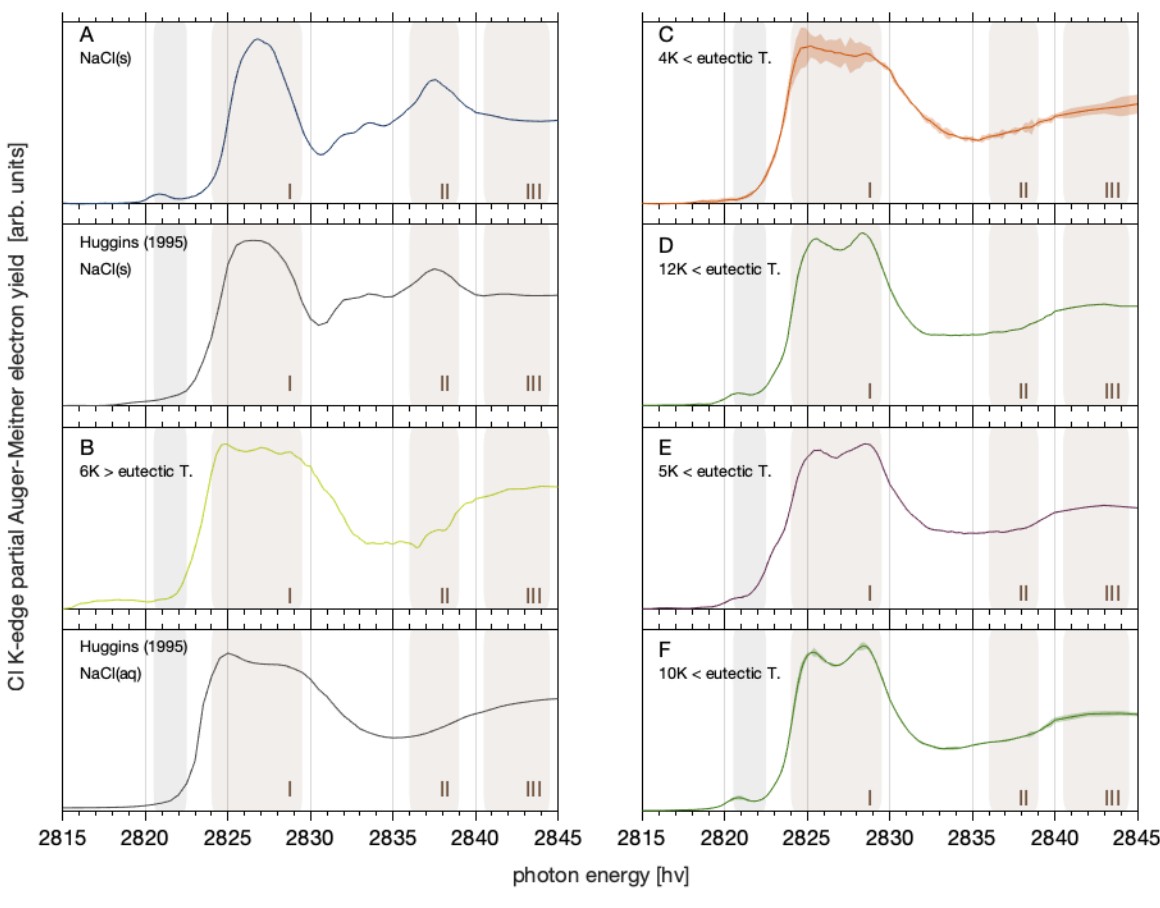


**Figure 3:** Partial electron yield chlorine K-edge NEXAFS spectra of the sodium chloride -- water binary system: **A** Solid NaCl at 248 K and 44 % relative humidity. **B** Aqueous NaCl solution in equilibrium with ice at 88 % RH and 259 K. **C** An averaged spectrum at the thermodynamic ice stability line at 248-249 K. **D** An individual spectrum upon further cooling to 240 K. **E** An individual spectrum upon heating back to 249 K in the ice stability domain. **F** The averaged spectrum at 244 K and relative humidities of 59 % and 73 %, lower than the ice stability domain. See Fig. 4 for precise measurement settings. The shaded area in the colour of the graph in C and F denote the standard deviation of 3 and 2 repeated NEXAFS acquisitions. Also shown are NEXAFS spectra of NaCl salt and aqueous solutions for comparison that were detected in fluorescence mode and not in partial electron yield (Huggins and Huffman, 1995). The brownish shaded area (I-III) highlights regions in the NEXAFS spectra disused in the text. The grey shaded area at 2821 eV highlights the photon energy region where carbon-chlorine bonds from carbon contamination might show an absorption feature (see text for details).







The spectrum of sodium chloride in aqueous solution, Fig. 3B, shows a broader absorption peak in region I compared to the
spectra of the halite and a second feature at 2840 eV (region III) (Huggins and Huffman, 1995). In this work, the NEXAFS
spectrum of NaCl in aqueous solution was recorded in a solution-ice binary mixture based on the phase diagram (see
Efflorescence at the Interface). Based on freezing point depression data, the concentration of sodium chloride in such an
aqueous solution in equilibrium with ice is 3.5 mol $l^{-1}$ (Rumble, 2019). The spectrum in Fig. 3B generally agrees with the X-
ray absorption spectra reported for 0.1 mol $l^{-1}$ and 1 mol $l^{-1}$ aqueous solutions (Fig. 3 Huggins and Huffman (1995)) as it
captures the general decrease in intensity in region I with excitation energy and the increase in absorption in region III (Fig.
3). Discussing differences in the hydration structure of chloride at the water—air interface as compared to the bulk solution is
beyond the scope of this work. Harada et al. (2011) has reported differences in the hydration structure of Br- at the aqueous
surface compared to in bulk. Due to the low spectrum quality, evident by the wiggles in region I, we refrain from discussing
the two distinct features in region I, at 2824 eV and 2829 eV, that have been observed in previous work on NEXAFS spectra
of aqueous chloride solution (Huggins and Huffman, 1995; Antalek et al., 2016).

Figure 3C-E show NEXAFS spectra acquired in the ice stability domain at temperatures below the eutectic temperature of
251.9 K (Koop et al., 2000a). At 248-249 K, the NEXAFS spectrum resembles that of a typical aqueous solution (Fig. 3C).
Figure 3C shows an average of 3 individual NEXAFS spectra which still shows substantial scatter that results in low spectra
quality. We assign this to the low amount of salt within the interfacial region and to potential thermal circulations and thus
redistribution of the liquid upon irradiation by the beam. Despite this uncertainty, the results in Fig. 3A-C allow to clearly
differentiate between solid halite and aqueous solution.

Upon cooling further to 241 K, the spectrum changed significantly. The NEXAFS spectrum in Fig. 3D shows a wide peak in
region I with two well resolved features about 3 eV apart. That both features have a similar intensity, makes this spectrum
clearly distinct from those of an aqueous NaCl solution with its decreasing trend of absorption in region I. Compared to the
spectrum of aqueous NaCl solution (Fig. 3B, C), the absorption edge is shifted to higher photon energies in the spectrum in
Fig. 3D. The absence of a feature in region II makes the spectrum in Fig. 3D distinct from the spectrum of anhydrous NaCl
salt. Notably, the spectrum quality is greatly improved and is similar to that of the solid, anhydrous halite sample (Fig. 3A).
One factor impacting the spectral quality is the stability of the sample during the NEXAFS acquisition. The analysis of the Cl
2p photoemission spectra acquired before and after each NEXAFS run showed that the amount of chlorine detected in the
sample volume fluctuated by less than 10%, between 0 % and 9 %, in samples shown in Fig. 3 C, D, E, and F (Appendix A).
For comparison, samples in Fig. 3 A and B showed a decrease of 39 % and 43 % in the integrated Cl 2p signal intensity,
respectively, from prior- to after the NEXAFS was recorded. Possible reasons for these trends are an increase in adsorbed
water with time masking the intensity of the underlying chlorine (Fig. 3A) and changes in the distance of the sample to the
electron analyser with time leading to a reduction in the intensities of all compounds (Fig. 3 B). Both of these processes would





also affect the NEXAFS signal by inducing changes in intensity with time. Direct quantitative comparison is beyond the scope
of this work, and is hampered by the different probing depth of XPS and NEXAFS as given by the kinetic energy.

We assign the spectrum in Fig. 3D to that of sodium chloride dihydrate (hydrohalite, NaCl•2H$_2$O). The NEXAFS spectrum of
the hydrohalite has to the best of our knowledge not been described before. The double peak feature in region I we observe
here is typical for other chloride hydrates such as MnCl$_2$•4H$_2$O, CaCl$_2$•2H$_2$O, and MgCl$_2$•6H$_2$O (Evans et al., 2008). It is also
present in the NEXAFS spectra of sodium chloride solution (Huggins and Huffman, 1995), with a different ratio of the two
features as observed here for the hydrohalite. Antalek et al. (2016) has explained this double peak feature in NEXAFS spectra
of aqueous chloride solutions by the formation of two distinct solvation shells with two distances between the chloride and the
solvating water molecules based on NEXAFS and extended X-ray absorption fine structure spectroscopy as well as molecular
dynamics simulations. We propose here that the NEXAFS spectra of the hydrohalite can be understood based on the same
argument, as the chloride in NaCl•2H$_2$O is coordinated by two sodium and four water molecules with differences in the distance
of each Cl to the neighbouring oxygen of the water molecules (Klewe et al., 1974).

Figures 3E and 3F show hydrohalite spectra after warming the sample back to 249 K in the presence of ice and at 244 K in
absence of ice. We'd like to note the excellent reproducibility of the spectra as seen in Fig. 3F by the small standard deviation
of two samples (shaded area). In summary, we have clearly presented three different chlorine K-edge NEXAFS spectra and
argued that these derive from the halite, chloride solution, and the hydrohalite. Hydrohalite at the air-ice interfacial region was
thus observed in the temperature range 241 K – 249 K.





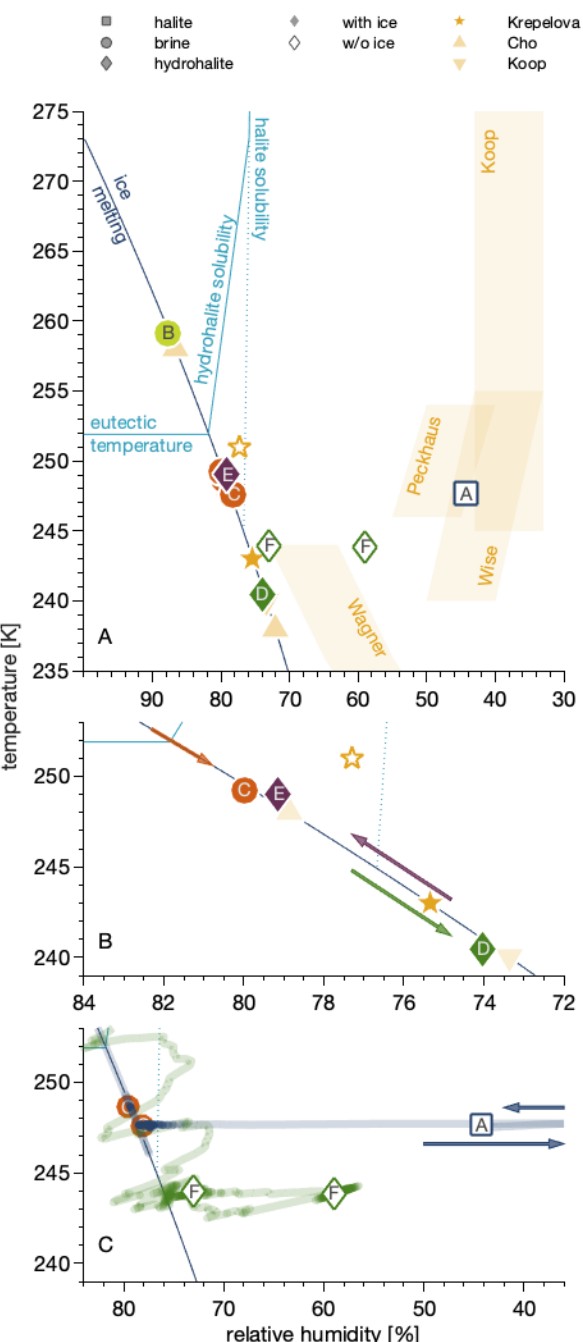

**Figure 4:** Phase diagram of the NaCl-water binary system showing the conditions and trajectories of the data in Fig. 3 in the relative humidity – temperature space. The relative humidity is the partial pressure of water relative to the vapour pressure of (supercooled) water as parameterised by Marti and Mauersberger (1993). The dark blue line denotes settings where ice is stable as the partial pressure of water matches the vapour pressure of ice. Above the eutectic temperature, it thus separates regions where ice and NaCl solution coexist from those where only NaCl solutions exist. The light blue lines denote conditions where anhydrous NaCl salt ('halite solubility') and where NaCl•2H$_2$O ('hydrohalite solubility') forms a solution when increasing the relative humidity at constant temperature; and where NaCl•2H$_2$O dissolves with increasing temperature at constant relative humidity ("eutectic temperature").

**A** Data from this work compared to previous results. Also shown are the data ranges where efflorescence has been observed in earlier work as shaded areas (Koop et al., 2000a; Wagner et al., 2011; Wise et al., 2012; Peckhaus et al., 2016). **B** Zoom into data below the eutectic temperature. The arrows show the sequence of changes in experimental temperature and relative humidity conditions. **C** Zoom to the data in absence of ice. The lines indicate the relative humidity and temperature trajectories to reach the experimental conditions for these data points. The arrows illustrate that the relative humidity was reduced to 0 % prior to returning to the conditions of the measurement.










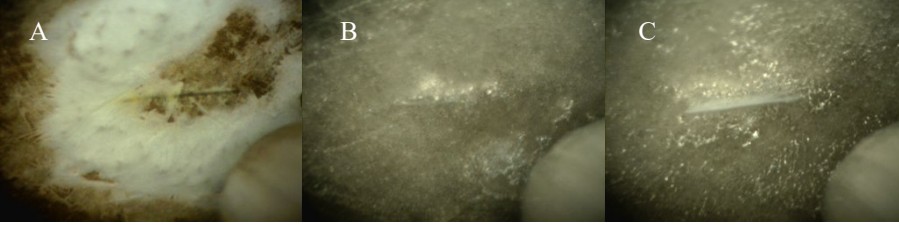


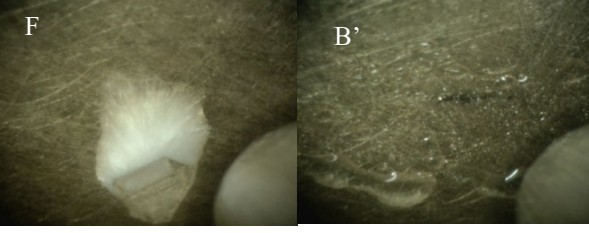

**Figure 5:** Optical microscopy pictures of the frozen samples. Each picture shows a 1 mm wide section of the sample holder with the sample.
The letters refer to the samples in Figs. 3 and 4. Picture B' shows the deliquesced sample prior to freezing.





### Efflorescence at the interface

Now that we have identified halite, the hydrohalites, and the aqueous solution by means of the NEXAFS spectra at the interfacial region, we discuss their observation in the phase diagram. Figure 4A shows the sodium chloride – water phase diagram in the temperature – relative humidity space as initially constructed by Koop et al. (2000a). In this work, the relative humidity (RH) is referenced to the vapour pressure of (supercooled) water as parameterised by Marti and Mauersberger (1993). There are two sets of experiments, those where ice is present and the sodium chloride is in equilibrium with ice (filled symbols) and experiments in absence of ice (open symbols) that were done with a relative humidity less than the ice stability line in the phase diagram (dark blue line). Generally, we have observed solid sodium chloride as halite or hydrohalite at temperatures below 240 K and at 44 % to 79 % RH and brine in the temperature range of 248 K to 259 K and RH of 78 % - 88 %.

### Liquid below eutectic and nucleation

In a typical experiment, anhydrous salt was exposed to increased relative humidity at a fixed temperature of 259 K. The relative humidity was increased by increasing the flux of water vapour into the experimental cell. Once the relative humidity reached 72 %, the sample started to dissolve by water up-take from the gas-phase and an aqueous solution was formed (brine). This phase change was evident by the sample becoming shiny and then forming transparent spheres as observed by an endoscope digital camera (Fig. 5B'). Then, the relative humidity was further increased and/or temperature was lowered to cross the ice stability line until ice nucleation occurred at a modest oversaturation. Ice nucleation was evident by a sharp pressure drop from the pressure dosed to the cell to the water vapor pressure of ice at that temperature. Generally, in the presence of ice, the partial pressure of water in the flow-through cell is given by the vapor pressure of ice and thus a sole function of temperature. If the water vapour pressure upstream of the flow through cell exceeds this value, the ice on the sample holder is growing, if it is set below, the ice sublimates. Based on the calibration of the dosing reservoir temperature and partial pressure of water in the flow-through cell (in absence of ice), the incoming $H_2O$ vapour flux was adjusted such that the equilibrium pressure in the cell matched the vapour pressure of ice.

A NEXAFS spectrum was acquired at 259 K and 1.82 mbar on the ice stability line and taken as reference for aqueous sodium chloride solution (Fig. 3B). When the temperature was lowered while adapting the flux of water into the set-up to match the vapour pressure of ice at 248 K - 249 K, thus 4 K to 5 K below the eutectic temperature (Fig. 4B), the NEXAFS spectra revealed that the chloride at the air-ice interface is in an environment identical to aqueous chloride (Fig. 3C). While cooling further and adjusting the vapor pressure to match that of ice at each temperature, a sudden change in the sample appearance, becoming less transparent, was observed by the digital endoscope, indicating efflorescence of the sample. A NEXAFS spectrum recorded reveals that hydrohalite has precipitated from the brine at 240.5 K and 74 % RH (Fig. 3D). Krepelova et al. (2010a) has studied phase changes of sodium chloride at the interfacial region of sodium chloride – water binary mixtures



previously. They have probed the oxygen with XPS and partial electron yield NEXAFS spectroscopy and concluded that, in
the presence of ice, hydrohalite forms about 11 K below the eutectic (filled star in Fig. 4). Consistent with that, the chloride
has a local environment indistinguishable from that of the hydrohalite 11.4 K below the eutectic temperature and in the
presence of ice in the current study. Our results of precipitation in the presence of ice surfaces agree with the crystallization
temperature observed by Koop et al. (2000a). We lack a direct comparison to the bulk, because the electron yield NEXAFS
spectroscopy used in our work is inherently surface sensitive. Cooling sodium chloride solutions of varying concentration,
Koop et al. (2000a) found precipitation of hydrohalite at 240 K in the presence of ice for bulk samples. Because precipitation
occurred at 20 K higher temperatures compared to emulsion samples of the same concentration, the authors concluded that the
presence of surfaces enhance the crystallization rate. In that study, the hydrohalite was identified by the melting temperatures
of the ice-hydrohalite mixtures. Malley et al. (2018) observed crystallization of sodium chloride solutions 1 K below the
eutectic temperature in the presence of ice. The hydrohalite was clearly identified using bulk sensitive Raman spectroscopy.
This difference in crystallization temperature may reflect the stochastic character of freezing, as already noted by Koop et al.
(2000a) when discussing the scatter in their data. The precise crystallization temperature is also influenced by freezing rate,
concentration, and the availability of surfaces (Bartels-Rausch et al., 2014). It appears thus that the precise occurrence of
crystallisation is governed by stochastics both at the surface and in heterogeneous freezing of the bulk.

To investigate whether or not precipitated sodium chloride is the stable form at the interface at temperatures close to the
eutectic temperature, the sample at 240 K (green filled triangle, Fig. 4B) was warmed towards the eutectic temperature while
staying in the ice stability domain. Acquiring a NEXAFS spectrum (Fig. 3E) that resembles that of hydrohalite, shows that the
solid salt is the thermodynamically stable form also at temperatures close to the eutectic. We therefore interpret the existence
of liquid during the previous cooling of the sample (Fig. 4B, orange circle, Fig. 3C) at the interface as supercooled solution.
The sample was kept at this condition for 3 h and showing that liquid can exist for extended times at the air-ice interface below
the eutectic temperature and that the temperature alone is not sufficient to predict its presence. Rather the thermal history of
the snow needs to be considered.

For samples that were cooled to temperatures that triggered efflorescence, the chlorine NEXAFS spectra show that the
hydrohalite is the dominating phase at the interface of frozen sodium chloride – water binary mixtures.  Cho et al. (2002) have
shown that when frozen aqueous solutions were warmed, a liquid fraction was observed below the eutectic temperatures.  In
their experiments, ice was frozen in NMR tubes lowering the temperature to 228 K in 15 min. which is significantly colder
than the efflorescence temperatures observed here and by Koop et al. (2000a). After 10 minutes, the samples were warmed
and NMR signals were recorded. Interestingly, Cho et al. (2002) have observed the liquid fraction only in experiments where
the sodium chloride concentration in the initial aqueous solution was below 0.01 mol l$^{-1}$. If the initial aqueous solution had a
concentration of 0.5 mol l$^{-1}$ no indication of liquid features below the eutectic were found. Tasaki et al. (2010) has shown a
similar concentration dependence for sodium bromide solutions using X-ray absorption reporting solvated bromide in the bulk





of the samples below the eutectic temperature only for concentrations below 50 mmol l$^{-1}$. The experiments described here
started with an aqueous solution that was formed in-situ and was kept in equilibrium with a vapour pressure of roughly
1.9 mbar. The chloride concentration in such solutions is close to the concentration in a solution at 1.8 mbar and at 259 K,
where ice nucleation occurred and where the freezing point depression data give a concentration of 3.5 mol l$^{-1}$. This
concentration can be directly compared to the concentration in the initial solutions of Cho et al. (2002), which ranged from
below 0.01 mol l$^{-1}$ to 0.5 mol l$^{-1}$. This back-of-the-envelope calculation thus suggests that the concentration of the solutions
from which ice nucleated in the experiments reported here exceeded those described by Cho et al. (2002) for which no liquid
fraction was observed. Now, the concentration of the initial solution from which ice precipitated, determines the ice to brine
ratio after ice formation. This is, because as the concentration of the brine is a sole function of temperature, the volume of the
brine relative to that of ice is given by the water to sodium chloride ratio in the initial solution. One might speculate that with
large amounts of brine relative to ice, that is concentrations of initial solutions from which ice nucleates > 0.5 mol l$^{-1}$, patches
and inclusions are larger in size than for more dilute solutions. The size of these patches or inclusions is of relevance, as surface
forces reduce the melting point only for inclusions in the nanometre range (Nye, 1991; Aristov et al., 1997; Christenson, 2001;
Bartels-Rausch et al., 2014). The absence of this inverse Köhler effect in larger particles might explain the lack of liquid
features both in the results reported here at the interface of ice and in the high concentration samples of Cho et al. (2002).
Support for large patches at the interface when solutions are frozen comes from microscopy studies. Low temperature scanning
electron microscopy work suggested the ice surface of frozen 0.05 mol l$^{-1}$ sodium chloride – water mixtures being covered by
μm sized brine features (Blackford, 2007; Blackford et al., 2007). Malley et al. (2018) used Raman microscopy of sodium
chloride solutions between 0.02 – 0.6 mol l$^{-1}$ initial concentration to identify micrometre-sized, partially connected patches of
liquid covering 11 % to 85 % of the ice surface at temperatures above the eutectic. Despite the impact of freezing temperature
and rate -- that differs among the individual studies -- on the distribution of impurities (Bartels-Rausch et al., 2014; Hullar and
Anastasio, 2016), these results clearly show the tendency of μm sized features dominating at the air-ice interface. In the
presence of nano-inclusions, we would also expect the deliquescence to occur at a lower temperature. This was not observed
in our experiments, suggesting the absence of nano-inclusions in the experiments presented here in the interfacial region.

We have recently reported chloride forming solvation shells in the interfacial region of ice upon adsorption of HCl at 253 K
(Kong et al., 2017). The surface concentration as derived from XPS suggested that it was done in the stability domain of ice,
i.e., the concentration of HCl was too low to melt the ice. Oxygen K-edge NEXAFS spectra showed that a substantial fraction
of the water molecules at the air-ice interface is arranged in a hydrogen-bonding structure like that of liquid water. The
NEXAFS spectrum of sodium chloride – ice mixtures (Fig. 3E) at 249 K are indistinguishable from those of hydrohalite. Taken
the spectra quality and the small difference in the shape of the liquid and of the hydrohalite spectrum, it is beyond the scope
of this work to elaborate whether the NEXAFS spectrum in Fig. 3E might be understood by deconvoluting it in its hydrohalite
and brine components and by this reveal a fraction of the chloride being embedded in a brine-like hydrogen bonding network.
Krepelova et al. (2010a)'s oxygen K-edge spectra of sodium chloride -- ice did not reveal water molecules being coordinated



like in the liquid. Taken together it seems likely that the chloride at the hydrohalite – air interface does not engage in forming
solvation shells similar to those in aqueous solutions and that the signal from the hydrohalite by far exceeds the signal from a
chloride that might migrate to the ice surfaces in intensity.

**Nucleation in absence of ice**

Hydrohalite can also precipitate in absence of ice by evaporating water from a solution at temperatures below 273 K (Craig et
al., 1975; Yang et al., 2017). Such a trajectory, that is the temperature and water vapor pressure the sample experienced, is
shown in Fig. 4C (green solid line).  Two NEXAFS spectra were recorded (Fig. 4, green diamonds), both of which were
identified as hydrohalite (Fig.  3F). The first one at 244 K and 59 % RH, where in absence of ice nucleation was visually
observed The location in the phase diagram is in agreement with Wagner et al. (2012)'s observation of salt deposits in aerosol
droplets in an aerosol chamber (AIDA)in absence of ice. The second NEXAFS spectra resembling that of the hydrohalite and
in absence of ice was recorded at a slightly higher relative humidity of 72 % (Fig.  4, green diamond). The sample at 44 % RH
(Fig. 4C, blue line) has been exposed to 0 % RH prior to acquiring the NEXAFS spectrum (Fig. 4, blue square), which removes
the crystal water (Light et al., 2009; Wise et al., 2012). This NEXAFS spectrum recorded at 248 K and 44 % RH shows that
no deliquescence occurred and thus serves as reference for a halite spectrum (Fig. 1A). We note that we misinterpreted this
spectrum as that of hydrohalite in our previous technical paper introducing the NAPP end station (Orlando et al., 2016).
The halite spectrum (Fig. 3A, Fig. 4C blue square) was recorded at 44 % RH, a humidity where water readily adsorbs on
surfaces (Bluhm, 2010), and the presence of water at the sodium chloride - air interface at elevated RH has been demonstrated
previously (Ewing, 2005; Wise et al., 2008). A further reason for the presence of water at lower RH than the deliquescence
might be sodium chloride's high solubility. Kong et al. (2020) has recently suggested that sodium ions from sodium acetate
are dissolved in adsorbed water prior to deliquescence. Indications for solvation of halide ions on single crystals below
deliquescence relative humidity were also reported previously (Luna et al., 1998). An analogue investigation was beyond the
scope of this work. The fact that we find no indication for a dominant liquid feature in the upper 6 nm of the sodium chloride
– air interface based on the chlorine X-ray absorption spectrum might be due to probing deeper into the interfacial region in
this work as compared to Kong et al. (2020).

**NaCl or NaCl.2H2O**

We have shown that in the presence of ice, hydrohalite precipitates; halite was not observed in this study except when crystal
water was removed in vacuum. This is in agreement with Koop et al. (2000a) who found precipitation of hydrohalite in the
presence of ice and suggested that heterogenous nucleation at ice surfaces favours nucleation of hydrohalite. Studies with
liquid droplets in absence of ice found both halite and hydrohalite precipitating in the thermodynamic stability domain of the
hydrohalite (Wagner et al., 2011; Wise et al., 2012; Peckhaus et al., 2016). The transition between crystallization of halite and
hydrohalite at between 253 K to 241 K was explained based on nucleation theory and the deviating trend of nucleation rate of



both species with temperature (Peckhaus et al., 2016). That hydrohalite crystallizes in absence of ice in our study shows that
also gold surfaces, of the sample holder, serve as good nucleation support for hydrohalite. It appears that any specific properties
of the ice surface, as compared to gold, are of minor importance. In agreement, Wagner et al. (2015) showed efficient
nucleation of hydrohalite in droplets containing solid oxalic acid and halite precipitated in absence of oxalic acid at the same
temperature and relative humidity.
**1 Summary and Atmospheric Implication**
The upper few nanometre of the interfacial region of a sodium chloride – ice binary system was investigated in this study at
various positions in the phase diagram. The sample was always in equilibrium with gas-phase water which makes cooling
conditions and concentrations of the brine identical to aerosol particles embedded in snow or in the troposphere.  The inherent
sensitivity to the interfacial region comes from using partial Auger-Meitner electron yield NEXAFS and the limited pathlength
of electrons travelling in matter. With a probing depth of a few nanometre, the interfacial region is probed spanning from the
upper molecular layers somewhat into the bulk (Ammann et al., 2018). In this work, we describe the NEXAFS spectrum of
hydrohalite for the first time.

The results emphasise that the nucleation of hydrohalite is a function of both the temperature and the relative humidity. While
we show that the nucleation of hydrohalite at the interface is favoured by surfaces, we find supercooled solution of sodium
chloride in the interfacial region of ice. The supercooled solutions have been observed to be metastable for hours in these
experiments. This has direct implications for heterogeneous chemistry in cold parts of Earth's environment. Multiphase
reactions may proceed at accelerated rates in these highly concentrated brines at temperatures ~10 degrees below the eutectic
compared to reactions on solid hydrohalite.  In this respect, there is no difference in the interfacial region to liquid embedded
in the bulk of snow and ice samples. This implies that when chemical reactivity of ice and snow is discussed, knowledge of its
thermal history is essential. From temperature and relative humidity alone, nucleation or efflorescence cannot be predicted at
the interface as in the bulk.

We suggest that the brine observed by Cho et al. (2002) is a consequence of micro-pockets. Because these are only observed
at low concentration of the solution before freezing of ice, but aerosol at typical relative humidity that prevail in cold parts of
the atmosphere are highly concentrated, we suggest that micro-pockets in the interfacial region small enough to establish a
significant depression in freezing point at the air-ice interface are of small relevance to the environment.



The appearance of hydrohalite at air-ice interfaces might also be of interest to sea ice research, because precipitation of
hydrohalite increases the albedo of sea ice. During the Snowball Earth period, 700 million years before present, climatic
conditions may have favoured the existence of hydrohalite with its climatic feedback (Light et al., 2009). For modern Earth,
precipitation of brine constituents in sea ice is relevant for ion mobility and might result in ion fractionation during wash out
events (Maus et al., 2008; Obbard et al., 2009; Maus et al., 2011).

Precipitation of sodium chloride in sea spray aerosol in the troposphere, not embedded in snow or sea ice, is further of ongoing
interest. The focus is placed on whether anhydrous sodium chloride (halite) or hydrohalite precipitates. In regions of the phase
diagram, where the hydrohalite is the thermodynamic stable form, precipitation of the halite was observed with impacts on
stability of the solid phase upon warming and or humidification, since the deliquesce relative humidity of the two compounds
differs by 6 percentage points (Wagner et al., 2012; Wise et al., 2012; Peckhaus et al., 2016).



## Data Availability

Bartels-Rausch, Thorsten (2020). Data set on interfacial supercooling and the precipitation of hydrohalite in frozen NaCl solutions by X-ray absorption spectroscopy. EnviDat. *doi:10.16904/envidat.164*.

## Acknowledgement

Funding by the Swiss National Science Foundation (SNSF) under Grant No. 149629 is acknowledged. We thank Mario Birrer (PSI) for his technical assistance. TB-R thanks Peter Alpert for stimulating discussion in our office.

## Author Contribution

TB-R designed and planned the study, analysed the data, and wrote the manuscript. FO, LA, MA, TH planned and optimised beamline and electron analyser settings. XK, AW, LA, and TBR performed the experiments. All authors approved the submitted version of the manuscript.

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




**Appendix A**




**Figure A1:** It shows the photo emission intensities as acquired before (blue line) and after (red line) each NEXAFS spectrum. The PE spectra of sodium (Na1s) are shown in column 1, of oxygen (O1s) in column 2, of chlorine (Cl2p) in column 3, and of carbon (C1s) in column 4. All spectra were acquired at the Phoenix beam line of SLS of PSI with a photon energy of 2200 eV. Pass energy was set to 100 eV and dwell time to 100 ms. Beam line slits were 2 x 2 mm. The C-H feature at 285 eV of the C1s





photoemission spectra served as binding energy reference and all spectra were shifted by 4-5.5 eV to account for charging effects.

The spectra in Fig. A1 can be described as follows:

- The Na1s region shows one gauss-shaped feature at 1072 eV as expected for sodium.
- The O1s region shows one dominant feature at 535 eV in line with oxygen in ice and a smaller feature at higher binding energy which might be attributed to gas-phase water.
- The Cl2p region shows the typical doublet of the p-orbital spin-orbit splitting (p(3/2) and p(1/2)), 1.6 eV apart at 200 eV binding energy. Additionally, in some spectra a small feature at 203 eV binding energy is evident which might be attributed to the Cl2p(1/2) of organic carbon species.
- The C1s region shows a broad spectrum with the main feature at 285 eV binding energy, typical for the C-H of adventitious carbon. The overlapping features at higher binding energy can be attributed to C-OH, C=O, C(O)=O and C-Cl.
- The spectra of all species acquired before and after the sample in Figure A are significantly wider compared to the other samples. Additionally, the Cl2p features are shifted by 2 eV to lower binding energy. This sample is, unlike the others, is characterised by a large variation in sample thickness (Figure 3A of manuscript) which leads to differential charging and might explain the wider peak shapes and shift in the Cl2p features.
- In sample A, the photoemission intensities of the Na1s and Cl2p show a higher intensity after the NEXAFS spectrum was acquired compared to before. During that time, the photoemission intensity of O1s increased. This suggests that water kept adsorbing to the sample and masked the signal intensity of the underlying sodium chloride.
- In sample B, all photoemission intensities decreased during the time it took to acquire the NEXAFS spectra. This suggests that the working distances changed which impacts the sensitivity of all sample components.