# Peer review of "Interfacial supercooling and the precipitation of hydrohalite in frozen"

_The Cryosphere, 2020_

## Referee Comment (RC1) · Subha Chakraborty (Referee) · 6 Jan 2021

The work from T. Bartels-Rausch et.al. titled, "Interfacial supercooling and the precipitation of hydrohalite in frozen NaCl solutions by X-ray absorption spectroscopy" demonstrates the first known NEXAFS studies of the interfacial phase transition properties of NaCl-H2O system at sub-freezing temperatures. The group has previously published significant original research and review articles on cryogenic atmospheric chemistry, including X-ray spectroscopy at the air-ice interface. This work adds to their NEXAFS research at the air-ice interface by showing the spectra of hydrohalites. The manuscript demonstrates a technique to identify phase transitions in frozen NaCl solution and a

method to observe chemistry in the first few nm of the surface. The manuscript, in my opinion, is well-communicated except for a few things, which I believe can be omitted for brevity and another couple of things requiring clarifications.

Some suggestions and corrections: 1. Fig. 3: the unit in the horizontal axis should read [eV], not [h$\nu$]. 2. The phase diagram of NaCl-H2O binary system is redundant as it has been investigated for years. However, the representation used in this manuscript is different from the conventional representation in terms of wt% or molal concentration. Here the authors used a molar concentration representation which must have taken into account volume contraction of the solution. However, it is difficult to find these in the articles they cited for reference. Koop 2000b has not directly shown the data shown in FIG. 2 of the manuscript. The CRC handbook 100th edition released only very recently. Although not a big concern, but I would suggest providing the molal labels in the x-axis as well along with the molar labels which are easier to trace for using the phase diagram. 3. Do you have an estimate of the cross-sectional area from which the spectra are being collected? Several groups showed physically separated ice and brine channels in frozen solutions in sub-100 $\mu$m scales (ACS Earth Space Chem. 2018, 2, 702$-$710, Langmuir 2014, 30, 5441$-$5447, Langmuir 2016, 32, 527$-$533, Cold Regions Science and Technology 138 (2017) 24–35). Does an average spectrum from a large area covering pure ice and brine have any effect on the intensity (and shape) of the spectra? 4. In the same line of thoughts, when hydrohalites are formed, they also cover a fraction of the ice surface (ACS Earth Space Chem. 2018, 2). Does it have an effect on the shape and subtility of the spectra? This might have direct implication on assigning the correct spectrum of the hydrohalite from FIG. 3D. How do you confirm that the spectrum in FIG. 3D is entirely from hydrohalite, and not from a mixture of liquid and solid phase co-existing as proposed by Cho et al. (J. Phys. Chem. B, Vol. 106, No. 43, 2002)? 5. In absence of NaCl, what would the spectra look like in the 2825 – 2830 eV ranges at different temperatures, knowing that these are the chlorine K-edge NEXAFS spectra? 6. In all cases, the authors started from a nearly 0.5 M.L-1 pre-frozen concentrations. Do the authors have any liquid spectra of

3.5 M.L-1 or higher concentrations along the liquidus line to check if the spectrum at a particular temperature down to Eutectic point is represented by brine at equilibrium at the given temperature? 7. Finally, the authors showed that down to 12 oC below Eutectic point, formation of hydrohalites are kinetically hindered. While Koop et al (J. Geophys. Res. 2000, 105, 26393) showed this is indeed possible down to 240 K, some other groups showed much lower hysteresis in their experiments (Phys. Chem. Chem. Phys., 2020,22, 17791-17797, ACS Earth Space Chem. 2018, 2). On the other hand, some results suggest that a little bit of contamination (surfactant-type) may also depress the formation of hydrohalite quite significantly (ACS Earth and Space Chemistry, 4 (2), 305ïÄ■310, (2020)). What do the authors believe that may lead to the large hysteresis?

---

## Referee Comment (RC2) · Anonymous Referee #2 · 15 Jan 2021

Bartels-Rausch et al present an elegant laboratory NEXAFS experiment investigating the presence of hydrohalite at the air-ice interface (top ∼6 nm) below the eutectic temperature, through aqueous NaCl experiments from 240 – 259 K. Notably, this work presents the first NEXAFS spectrum of hydrohalite through the advantage of probing the chlorine K-edge, in comparison to their previous work examining the oxygen NEXAFS spectra. Overall, the manuscript is well-written and has good motivation, particularly to air-snow interactions. Here, I present suggestions to improve the clarity of the manuscript and relevance to other previous work that further shows the utility of the current work.

Introduction: When discussing the links to air-snow reactions on Lines 34-49 and 67-82, it would be helpful to briefly discuss all of the reaction pathways shown in Figure 1. Regardless of the depth of discussion, references need to be provided in the figure caption, or at least in the main text, for the reactions shown in Figure 1. The authors discuss gas-phase OH reacting with chloride, but don't discuss aqueous OH reacting with chloride at the ice surface (Halfacre et al. 2019, Atmos. Chem. Phys.), which would seem to be of relevance. Of particular relevance, and not currently cited in this manuscript, is the work by Wren et al (2013, Atmos. Chem. Phys.) and Custard et al. (2017, ACS Earth & Space Chem.) that showed reduced snow/ice Cl2 production, in the lab and field, respectively, at temperatures below the eutectic, which was attributed this to the presence of hydrohalite, thereby directly connecting to this present lab study. Similarly, the lack of observed Cl2 at lower temperatures by Sjostedt and Abbatt (2008, Environ. Res. Lett.) was attributed to the presence of either halite or hydrohalite. Also, Lopez-Hilfiker et al. 2012 (Atmos. Chem. Phys.) also invoked the presence of hydrohalite to explain the relative production of ClNO2 vs Br2 in N2O5 reactions on saline ice.

Figure 5: Are optical images available for D-F as well? It would be particularly useful to refer to this, for example on Line 366 when the optical image is being described for D, for example. Also, please define the letters in the caption of Fig 5 so that reader is not required to refer back to Fig 3. In addition, consider changing the font on the letters and making them bold so that they are easier to discern; in particular, B is difficult to distinguish from B'.

Additional Comments: - Fix section numbering throughout (all start with 1)

- Lines 37-38, 41, 44-45, 80-82: Please add references to these sentences.

- Lines 57-65: This paragraph about bromine chemistry detracts from the focus of the current study and is suggested to be removed. Instead, it would be better to discuss the reaction mechanisms pertinent to chlorine chemistry shown in Figure 1.

- Figure 2: Please add the year to both Rumble citations in the caption and fix the spelling of "aqueous" in multiple locations the figure.

- Line 228: Fix typo "disused".

- Figure 4: Consider making the phase labeling on plot A more similar to Fig 2. For example, the labeling of the "ice melting" line was confusing at first given the locations of the individual words surrounding the line in the figure.

- Lines 348-359: Consider moving this paragraph to the methods section, as it describes how the experiment was conducted, rather than the results of the experiment.

- Line 370: Please provide the temperature here in parentheses for clarity (rather than just simply 11 K below the eutectic) to aid the reader in referring to Fig 4 and quickly finding the proper star marker.

- Line 371: Where is this "11.4 K below the eutectic" data shown? This sentence seems like it is discussing the current work, but Fig 3 only shows 10 K and 12 K below the eutectic.

- Lines 378-383: Is it possible that the hydrohalite may form within the bulk prior to the surface, explaining the higher temperature observed by Malley et al (2018) compared to this work?

- Line 387: For clarity, I suggest adding "at 5 K below the eutectic temperature" after "spectrum" in this sentence.

- Line 392: Change "snow" to "ice" here, since authentic snow was not studied in this work.

- Lines 394-426: This is a very long paragraph. Please consider breaking up.

- Lines 428 – 439: This paragraph overall should be revised to make it clearer and easier to read and relate the previous work to the current results. In particular, the goal of this paragraph could be clarified at the beginning of the paragraph to help guide the

reader, as I had to read the beginning sentences multiple times to understand them in the context of the current work.

- Lines 444-445: This sentence appears to be missing its end.

- Line 450: I believe the authors mean to refer to Fig 3A here.

- Line 475: Please provide an estimate or approximate range here in parentheses to provide improved understanding of what "the upper few nanometre" mean.

- Line 447: I suggest changing "identical" to "similar" here, as the sea salt aerosol in the environment are more complex than simple NaCl-H2O systems. Of particular relevance is that sea spray aerosol particles can have thick organic coatings (e.g., Kirpes et al. 2019, ACS Central Science).

- Lines 489 – 490: It would be useful to add discussion about the temperature ranges that are important to consider here (that would matter) when considering the polar environment that is being discussed in which temperature swings regularly occur with changing weather. A more detailed discussion referring to temperature ranges would be helpful to bring the gap to observations, based on the temperature and RH-dependent results of the current work.

- Lines 495-496: This discussion of micro-pockets is confusing when comparing to text on Line 424. Please clarify.

- Lines 500-501: Would the concentration effect discussed on page 17 have an impact here?

- Lines 503-504: Perhaps this would also have an impact on brine migration upward through the snow (i.e. Domine et al. 2004, Atmos. Chem. Phys.)?

- Lines 506-510: It would be useful to add discussion here about where in the atmosphere this might matter (using the temperature and RH knowledge from this work). Also, would the history of the production of sea spray aerosol as droplets (and therefore not starting at 0% RH) matter in terms of halite vs hydrohalite based on the results presented herein? Also, might the presence of organics coating the sea salt aerosol have an impact?

---

## Referee Comment (RC3) · Anonymous Referee #3 · 19 Jan 2021

Bartels-Rausch et al. present an original piece of research, investigating the phase changes in the NaCl-water binary system by measuring NEXAFS spectra across the chlorine K-edge when varying the temperature- and relative humidity conditions the particles are exposed to. The assignment of the three basic spectra types recorded for anhydrous NaCl particles, aqueous NaCl solution droplets, and the first-time measurement for hydrohalite, are convincing. I do not have any major concerns regarding the scientific content and quality of the manuscript, but I would strongly encourage the authors to improve to some extent the way of presentation of the data, and thereby better "guide the reader" through the various parts of the article.

Major comment:

All the while reading the section "NEXAFS of brine, halite, and hydrohalite", I kept wondering how exactly the experiment was carried out, i.e., what was the actual trajectory in the T-RH space, how exactly was the freezing of the NaCl solution droplets induced, how did you move along the ice/NaCl(aq) equilibrium line, how was hydrohalite without ice (Fig. 3F) formed? All these experimental details are only provided in later parts of the article (e.g. line 348ff, line 442ff). I see some good reason for the chosen manuscript structure, i.e., that you first want to describe the NEXAFS spectra, discuss some technical aspects like spectra quality and reproducibility, compare your measurements with literature spectra - and then later discuss the exact formation conditions for hydrohalite and the atmospheric implications based on the T-RH trajectory. But having a better general idea of the experimental procedure before reading the section with the NEXAFS spectra would be in my opinion a clear improvement regarding the clarity of presentation. One suggestion would be the following:

On line 84, you introduce the NaCl-water phase diagram, but describe it with some "hypothetical" trajectory, starting from a sample below 251.9 K and then increasing the temperature. But why not discuss the phase diagram with a "proper" trajectory from your experiments, which could be schematically depicted in Fig. 2 – meaning you start with an aqueous NaCl solution droplet, induce some supercooling to nucleate ice, and then move along the liquidus curve towards the eutectic and below, to investigate at which point hydrohalite precipitates. This is also the trajectory during which most of the spectra shown in Fig. 3 (B − E) were recorded (apart from the anhydrous NaCl, A, and the "ice-free" hydrohalite, F. At the end of the introduction or at the beginning of the "NEXAFS of brine, halite, and hydrohalite" section, you should then include a paragraph and inform the reader about the general structure of the manuscript, i.e., that you want to disentangle the detailed description of the NEXAFS spectra of the three species (shown first) from the detailed analysis of the phase changes when moving in the T-RH space (shown later). Please also number all section and subsection headings

correctly.

Additional comments:

1) Regarding line 88 and 112: Isn't 251.9 K the ice-hydrohalite eutectic, and shouldn't ice and NaCl*2H2O be the energetically favored phases below that temperature?

2) Line 279/280: "249 K in the presence of ice and at 244 K in the absence of ice": When reading this sentence for the first time, I was also wondering whether spectra 3E and 3F were from the same trajectory and asked myself how the experimental procedure could have been – it is only explained much later (line 441ff). You should include here at least a short description of how the particles from spectrum F were generated.

3) Line 337: Here starts the detailed discussion of the phase behavior of the particles in the T-RH space. I would also appreciate an introductory paragraph describing how this section is structured and what different aspects are discussed. Otherwise, the reader may quickly lose track of things. For example, the heading "Liquid below eutectic and nucleation" in line 347 comprises a very long section that could be divided into various subsections. In the headings, you could also be more specific what you mean by "nucleation", nucleation of ice or the precipitation of hydrohalite.

4) Line 353: Can you please quantify "modest" supersaturation – did you need to reach the homogeneous freezing limit for aqueous solution droplets (Koop et al., 2000b) or did the surface catalyze heterogeneous ice nucleation?

5) Line 442, regarding the trajectory when recording spectrum 3F: Could you please elaborate a bit more on the idea and temporal order behind this trajectory (Fig. 4C), I did not quite understand how the procedure was – did you again try to cool a NaCl droplet along the liquidus curve but without inducing sufficient supersaturation to nucleate ice? And then at about 244 K reduced the RH to 59% to induce the crystallization of hydrohalite? And then increased RH back to 73%?
Technical corrections:

1) Doesn't the title sound a bit awkward? Maybe better: "Investigation of interfacial supercooling . . ." or ". . . NaCl solutions studied by X-ray absorption spectroscopy"

2) Line 86: "shows a part of the phase diagram"

3) Line 109: "in equilibrium with ice"

4) Line 115 – 117: Very long sentence, please split into two.

5) Line 138: "nitrate and chloride form solvation cells"

6) Line 173: "and with a pass energy"

7) Line 186: "take-off angle of detected electrons (?)"

8) Line 220: photon energy in eV

9) Line 240: Br- (superscript is missing)

10) Line 338: "the hydrohalite"

11) Line 353: maybe better "supersaturation"

12) Line 355: maybe: "and is thus a sole function"

13) Line 411/412: This is a pretty nested sentence, please re-phrase.

14) Line 416: "in larger patches/inclusions"

15) Line 444/445: The sentence seems incomplete, it also misses the point at the end.

16) Line 450: Fig. 3A

17) Line 462: check super- and subscripts

---

## Author Comment (AC1) · 19 Feb 2021

Referee #1

The work from T. Bartels-Rausch et.al. titled, "Interfacial supercooling and the precipitation of hydrohalite in frozen NaCl solutions by X-ray absorption spectroscopy" demonstrates the first known NEXAFS studies of the interfacial phase transition properties of NaCl-H2O system at sub-freezing temperatures. The group has previously published significant original research and review articles on cryogenic atmospheric chemistry, including X-ray spectroscopy at the air-ice interface. This work adds to their NEXAFS research at the air-ice interface by showing the spectra of hydrohalites. The manuscript demonstrates a technique to identify phase transitions in frozen NaCl solution and a method to observe chemistry in the first few nm of the surface. The manuscript, in my opinion, is well-communicated except for a few things, which I believe can be omitted for brevity and another couple of things requiring clarifications.

*We thank Subha Chakraborty for the detailed comment and for the kind acknowledgement of our work in the field.*

Some suggestions and corrections:

1. Fig. 3: the unit in the horizontal axis should read [eV], not [hν].

*Thanks, done.*

2. The phase diagram of NaCl-H2O binary system is redundant as it has been investigated for years. However, the representation used in this manuscript is different from the conventional representation in terms of wt% or molal concentration. Here the authors used a molar concentration representation which must have taken into account volume contraction of the solution. However, it is difficult to find these in the articles they cited for reference. Koop 2000b has not directly shown the data shown in FIG. 2 of the manuscript. The CRC handbook 100th edition released only very recently. Although not a big concern, but I would suggest providing the molal labels in the x-axis as well along with the molar labels which are easier to trace for using the phase diagram.

*Figure 2 is certainly redundant; we agree with the referee. We added this purely to remind the reader and to initiate the introduction of Fig. 4 – the phase diagram in the relative humidity space – with this more common representation of the phase diagram. We will add a description of a typical experiment to introduce the concept of sample handling and preparation earlier based on the suggestion of another referee. And yes, you spotted our deviation in Fig. 2 from the classical phase diagrams in wt-%. The "Handbook" lists the freezing point depression data both in molarity and in molality and we preferred to use molarity for reasons of consistency as concentrations throughout the text are given in molarity.*

[Figure]

"*The focus of this work was to experimentally observe phase changes of sodium chloride below the eutectic temperature. A typical experimental procedure started with a dry sample of anhydrous sodium chloride (halite, NaCl) which was exposed to increasing gas-phase water at constant temperature of 259 K. By absorbing water from the surrounding air, a phase transition from the solid salt to a liquid solution (deliquescence) took place. Upon increasing the gas-phase water dosing further (Fig. 2, red arrow) ice crystalised and a two-phase system of ice and brine occurred (Fig. 2, red cross). After probing the sample at this position in the phase diagram (see below), temperature was lowered and the dosing of the water-vapor adopted to move along the liquidus line to below the eutectic temperature to perform additional measurements. During this cooling period, salt concentration and volume of the brine changes. Such changes with varying relative humidity (hygroscopic growth) have long been discussed for aerosol in the troposphere.*"

*To limit the number of units in this work we prefer to keep the units in Fig. 2 as they are. To give the reader better access to the data, we added a table to Appendix B listing the freezing point depression, molarity, and molality as given in the "Handbook" (Rumble, 2019):*

| mass fraction [%] | molal concentration [mol kg-1] | molar concentration [mol l-1] | freezing point depression [K] |
|---|---|---|---|
| 0.1 | 0.017 | 0.017 | 0.06 |
| 0.2 | 0.034 | 0.034 | 0.12 |
| 0.3 | 0.051 | 0.051 | 0.18 |
| 0.4 | 0.069 | 0.069 | 0.24 |
| 0.5 | 0.086 | 0.086 | 0.3 |
| 1 | 0.173 | 0.172 | 0.59 |
| 1.5 | 0.261 | 0.259 | 0.89 |
| 2 | 0.349 | 0.346 | 1.19 |

| | | | |
|---|---|---|---|
| 2.5 | 0.439 | 0.435 | 1.49 |
| 3 | 0.529 | 0.523 | 1.79 |
| 3.5 | 0.621 | 0.613 | 2.1 |
| 4 | 0.713 | 0.703 | 2.41 |
| 4.5 | 0.806 | 0.793 | 2.73 |
| 5 | 0.901 | 0.885 | 3.05 |
| 6 | 1.092 | 1.069 | 3.7 |
| 7 | 1.288 | 1.256 | 4.38 |
| 8 | 1.488 | 1.445 | 5.08 |
| 9 | 1.692 | 1.637 | 5.81 |
| 10 | 1.901 | 1.832 | 6.56 |
| 11 | 2.115 | 2.029 | 7.35 |
| 12 | 2.333 | 2.229 | 8.18 |
| 13 | 2.557 | 2.432 | 9.04 |
| 14 | 2.785 | 2.637 | 9.94 |
| 15 | 3.02 | 2.845 | 10.89 |
| 16 | 3.259 | 3.056 | 11.89 |
| 17 | 3.505 | 3.27 | 12.94 |
| 18 | 3.756 | 3.486 | 14.04 |
| 19 | 4.014 | 3.706 | 15.22 |
| 20 | 4.278 | 3.928 | 16.46 |
| 21 | 4.548 | 4.153 | 17.78 |
| 22 | 4.826 | 4.382 | 19.18 |
| 23 | 5.111 | 4.613 | 20.67 |

3. Do you have an estimate of the cross-sectional area from which the spectra are being collected?

*Thank you very much for this question. Indeed, an interesting parameter to include to facilitate comparison with previous studies. In our set-up this area is determined by the area from which electrons will reach the analyser, and not from the area that is exposed to X-ray. On first approximation, this electron acceptance area is given by the diameter of the analyser's sample orifice which is 500 μm in this work.*

*«The distance of the sample to the electron analyser inlet (working distance) was 1 mm. The electron analyser was operated with an electron sampling aperture with a diameter of 500 μm, which results in sampling roughly an area with a diameter of 500 μm of the sample from which the emitted electrons reach the detector.»*

Several groups showed physically separated ice and brine channels in frozen solutions in sub-100 μm scales (ACS Earth Space Chem. 2018, 2, 702–710, Langmuir 2014, 30, 5441–5447, Langmuir 2016, 32, 527–533, Cold Regions Science and Technology 138 (2017) 24–35). Does an average spectrum from a large area covering pure ice and brine have any effect on the intensity (and shape) of the spectra?
4. In the same line of thoughts, when hydrohalites are formed, they also cover a fraction of the ice surface (ACS Earth Space Chem. 2018, 2). Does it have an effect on the shape and subtility of

the spectra? This might have direct implication on assigning the correct spectrum of the hydrohalite from FIG. 3D. How do you confirm that the spectrum in FIG. 3D is entirely from hydrohalite, and not from a mixture of liquid and solid phase co-existing as proposed by Cho et al. (J. Phys. Chem. B, Vol. 106, No. 43, 2002)?

*Thank you for asking for clarification of the issue of the location of the brine within the frozen samples. Before answering the question, let me summarize that the X-ray absorption spectra that we present (Figure 3) probe exclusively chlorine. We (and others) have shown, that these spectra are sensitive to phase changes. In other words, the spectra intensities reflect the chlorine present in the samples and are insensitive to the fraction of ice. Sampling a larger area and thus more chlorine in the sample would indeed change the intensity of the observed spectra. Please note that the spectra in Fig. 3 are normalised to focus on and to compare their shape and any intensity information is lost. We can modify our set-up and work with either a 500 μm or 300 μm orifice, but we have not done this in this work.*

*Concerning the location of the hydrohalites (and brine). We agree with the referee that based on previous studies and on thermodynamic considerations a fractionation of brine and ice phases or hydrohalite and ice is very likely. The brine/hydrohalite can be present in channels, patches, micropockets. In this work, we focused on phase changes and the goal was to report the significant and large differences in the spectra of hydrohalite and brine. More subtitle changes of X-ray absorption spectra are known and have been reported with concentration of solutions, for example for chloride containing solutions. Investigating these, or differences of the X-ray absorption spectra for brine/hydrohalite at different locations, was beyond the scope of this work and would require different approaches such as a liquid-jet set-up. We happily include the suggested references and also address the question of the sample area when discussing the patches and nanopockets:*

*«Support for large patches at the interface when solutions are frozen comes from a number of studies {Malley, 2018; Krausko, 2014; Tokumasu, 2016; Lieb-Lappen, 2017}. Low temperature scanning electron microscopy work suggested the ice surface of frozen 0.05 mol l-1 sodium chloride – water mixtures being covered by μm sized brine features (Blackford, 2007; Blackford et al., 2007). Malley et al. (2018) used Raman microscopy of sodium chloride solutions between 0.02 – 0.6 mol l-1 initial concentration to identify micrometre-sized, partially connected patches of liquid covering 11 % to 85 % of the ice surface at temperatures above the eutectic. Despite the impact of freezing temperature and rate -- that differs among the individual studies -- on the distribution of impurities (Bartels-Rausch et al., 2014; Hullar and Anastasio, 2016), these results clearly show the tendency of μm sized features dominating at the air-ice interface. In the dominant presence of nano-inclusions, we would also expect the deliquescence to occur at a lower temperature. This was not observed in our experiments, suggesting the absence of nano-inclusions in the experiments presented here in the interfacial region. Please note, that the NEXAFS spectroscopy presented here probes an area at the interface of the sample with a diameter of about 500 μm. As the spectroscopy is selective to chlorine, we have no information about the fraction of brine versus ice in the probed part of the sample.»*

*How certain are we that spectrum 3D is not in fact a combination of brine and solid chloride phase? Well, we can't exclude the presence of small amounts of brine in neither sample (D and E) based on the X-ray absorption spectra. In the manuscript, we tried to argue for the*

*existence of hydrohalite as main phase rather than the absence of brine. We will carefully reword the manuscript to make this clearer. Thank you for pinpointing this shortcoming. Further, we will add a paragraph explicitly mentioning the possibility of small amounts of liquid. This is fully consistent with the current argumentation and conclusion as one might expect micropockets to show a size distribution resulting in a small fraction of pockets being small enough to stabilize liquid at a given temperature.*

*Indeed, spectra D and E show a small increase in intensity starting at 2823 eV which could be consistent with a contribution of liquid brine as spectrum B (brine) shows such a feature, but not spectrum F (hydrohalite). The spectrum shown in 3F was derived in absence of ice and at a partial pressure of water where brine -even at very high concentration- is not stable. The following graph shows results from a linear combination of spectrum F and spectrum B and a comparisons of the resulting spectrum to spectrum D (left graph) and E (right graph).*

[Figure]

*A linear combination of 10% B (brine) and 90% F (hydrohalite) reproduces the spectrum D indeed quite well. This clearly illustrates that we cannot rule out small amounts of brine in that sample. For spectrum E the situation is different. If we attempt to match the intensity of the features at 2825 eV and 2829 eV, a combination of 60% F and 33% E gives best results. However, this linear combination does neither match spectrum E at 2823 eV, nor at around 2835 eV. We assign this mainly to an insufficient quality of spectrum B (as detailed in the manuscript) and therefore prefer not to present this analysis in the manuscript. However, we will mention the possibility of small amounts of liquid brine and that the spectra do not rule this option out. Thank you for pointing this out and apologies for appearing so black and white.*

*We will update the discussion of the manuscript to make this clearer:*

*«The previous argumentation is based on the features in the NEXAFS spectrum of sodium chloride – ice mixtures shown in Fig. 3 D and E being dominated by the NEXAFS spectrum of hydrohalite shown in Fig. 3F. In particular the spectrum in Fig. 3E, acquired 3 K below the eutectic temperature, shows a shoulder starting at 2823 eV. Such a feature is absent in the spectrum of the hydrohalite (Fig. 3 F), but the spectrum of brine (Fig. 3 B) shows an increase in absorption starting at this X-ray energy. We can thus not exclude the presence of brine in the samples where the hydrohalite dominates the NEXAFS. Taken the spectra quality and the small difference in the shape of the liquid and of the hydrohalite spectrum, it is beyond the scope of this work to elaborate whether the NEXAFS spectrum in Fig. 3E might be understood by deconvoluting it in its hydrohalite and brine components and by this reveal a fraction of the chloride being embedded in a brine-like hydrogen bonding network. Two*

*reasons might explain the presence of liquid in these samples at sub-eutectic temperatures. First, one might expect a certain distribution in the size of micropockets and a small fraction of the pockets might thus be small enough to stabilize liquid at these temperatures. Secondly,…»*

5. In absence of NaCl, what would the spectra look like in the 2825 – 2830 eV ranges at different temperatures, knowing that these are the chlorine K-edge NEXAFS spectra?

*Because beamtime is rare and thus expensive, we have not recorded a Cl K-edge NEXAFS on a NaCl free surface. One would probably sample traces of chlorine impurities, but I question those being intense enough to give a relevant and significant signal. Note that this study was done with molar quantities of chloride.*

6. In all cases, the authors started from a nearly 0.5 M.L-1 pre-frozen concentrations. Do the authors have any liquid spectra of 3.5 M.L-1 or higher concentrations along the liquidus line to check if the spectrum at a particular temperature down to Eutectic point is represented by brine at equilibrium at the given temperature?

*Well, we started in all experiments with dry NaCl as water was pumped away in all samples when introducing the sample into the experimental set-up. Water was then dosed from the gas-phase forming brine and ice once the RH was sufficient high. The sample in our set-up is mounted vertically, which limits the possibility to sample liquids. In this work, the trick was to stabilize the liquid brine with the ice matrix by probing brine-ice binary mixtures. The only spectrum we have is that of brine at 259 K, the equilibrium concentration of which is 3.5 mol $l^{-1}$. To make this clearer, we have added a paragraph to the introduction when discussion the phase diagram there and modified Figure 2:*

[Figure]

*Figure 2: Phase diagram of the NaCl-water binary system. The data show the freezing point depression of sodium-chloride solutions (yellow filled circles) and give the concentration of an aqueous sodium chloride solution in equilibrium with ice in the temperature range of 273 K to*

254 K (Rumble, 2019). The dark blue lines indicate the phase boundaries (Koop et al., 2000b; Rumble, 2019), that is it denotes the so-called liquidus and solidus line, respectively, and thus shows the temperature and concentration range where ice and aqueous sodium chloride solution co-exist. The eutectic temperature of sodium chloride – water binaries is 251.9 K (Koop et al., 2000a). Also shown is a typical experimental procedure (red arrows and cross).

«The focus of this work was to experimentally observe phase changes of sodium chloride below the eutectic temperature. A typical experimental procedure started with a dry sample of anhydrous sodium chloride (halite, NaCl) which was exposed to increasing gas-phase water at constant temperature of 259 K. By absorbing water from the surrounding air, a phase transition from the solid salt to a liquid solution (deliquescence) took place. Upon increasing the gas-phase water dosing further (Fig. 2, red arrow) ice crystalised and a two-phase system of ice and brine occurred (Fig. 2, red cross). After probing the sample at this position in the phase diagram (see below), temperature was lowered and the dosing of the water-vapor adopted to move along the liquidus line to below the eutectic temperature to perform additional measurements. During this cooling period, salt concentration *and volume of the brine changes. Such changes with varying relative humidity (hygroscopic growth) have long been discussed for aerosol in the troposphere.*»

7. Finally, the authors showed that down to 12 oC below Eutectic point, formation of hydrohalites are kinetically hindered. While Koop et al (J. Geophys. Res. 2000, 105, 26393) showed this is indeed possible down to 240 K, some other groups showed much lower hysteresis in their experiments (Phys. Chem. Chem. Phys., 2020,22, 17791-17797, ACS Earth Space Chem. 2018, 2). On the other hand, some results suggest that a little bit of contamination (surfactant-type) may also depress the formation of hydrohalite quite significantly (ACS Earth and Space Chemistry,4(2),305˘310,(2020)).What do  the authors believe that may lead to the large hysteresis?

*We like to stress that nucleation is a stochastic process and therefore variation in the freezing point are expected. Further, freezing rate and amount of salt will play a role: Here we come back to the location of the brine. Depending on the concentration the size of brine batches varies and thus the formation of nano-pockets with lower freezing points due to surface curvature of the pockets is more or less likely. We address this now in an expanded paragraph and hope this is clearer:*

*This difference in crystallization temperature may reflect the stochastic character of freezing, as already noted by Koop et al. (2000a) when discussing the scatter in their data. The precise crystallization temperature is also influenced by freezing rate, concentration, and the availability of surfaces (Bartels-Rausch et al., 2014). It appears thus that the precise occurrence of crystallisation is governed by stochastics at the surface as has been shown for freezing of bulk samples (Alpert and Knopf, 2016). Because of the good agreement between the precipitation temperatures observed in this study and in (Koop et al., 2000a), we believe that the deviation from (Malley et al., 2018)'s results does not indicate differences in the freezing behaviour at the surface vs. in the bulk.*

*We judge the concentration of humic acid used in Chakraborty too high to explain the hysteresis in the data by contamination. We have added this study when discussing the impact of organics on freezing:*

*We suggest that further studies focus on samples with more complex chemical composition to enhance our knowledge of environmental multiphase chemistry. For example, organic compounds are a common constituent of sea-salt aerosol (O'Dowd et al., 2004) { Kirpes, 2019} and recently we have shown how there presence impacts the microphysics and thus reactivity of salt particles towards ozone (Edebeli et al., 2019). Further, {Chakraborty, 2020} has shown a depression in hydrohalite precipitation temperature in humic acid – sodium chloride mixtures.*

---

## Author Comment (AC2) · 19 Feb 2021

Bartels-Rausch et al present an elegant laboratory NEXAFS experiment investigating the presence of hydrohalite at the air-ice interface (top ∼6 nm) below the eutectic temperature, through aqueous NaCl experiments from 240 – 259 K. Notably, this work presents the first NEXAFS spectrum of hydrohalite through the advantage of probing the chlorine K-edge, in comparison to their previous work examining the oxygen NEXAFS spectra. Overall, the manuscript is well-written and has good motivation, particularly to air-snow interactions. Here, I present suggestions to improve the clarity of the manuscript and relevance to other previous work that further shows the utility of the current work.

*We thank the referee for the kind judgement.*

Introduction: When discussing the links to air-snow reactions on Lines 34-49 and 67- 82, it would be helpful to briefly discuss all of the reaction pathways shown in Figure 1. Regardless of the depth of discussion, references need to be provided in the figure caption, or at least in the main text, for the reactions shown in Figure 1. The authors discuss gas-phase OH reacting with chloride, but don't discuss aqueous OH reacting with chloride at the ice surface (Halfacre et al. 2019, Atmos. Chem. Phys.), which would seem to be of relevance. Of particular relevance, and not currently cited in this manuscript, is the work by Wren et al (2013, Atmos. Chem. Phys.) and Custard et al. (2017, ACS Earth & Space Chem.) that showed reduced snow/ice $Cl_2$ production, in the lab and field, respectively, at temperatures below the eutectic, which was attributed this to the presence of hydrohalite, thereby directly connecting to this present lab study. Similarly, the lack of observed $Cl_2$ at lower temperatures by Sjostedt and Abbatt (2008, Environ. Res. Lett.) was attributed to the presence of either halite or hydrohalite. Also, Lopez-Hilfiker et al. 2012 (Atmos. Chem. Phys.) also invoked the presence of hydrohalite to explain the relative production of $ClNO_2$ vs $Br_2$ in $N_2O_5$ reactions on saline ice.

*Apologies for our obviously too limited literature search. As excuse we might only say that we had expanded this discussion rather late in the writing process and maybe therefore not with the care necessarily. Thank you for pointing us to this reference, we will be happy to include a discussion on this work in a revised manuscript.*

*"In the cryosphere, where the snowpack is strongly impacting the chemistry in the overlaying atmosphere (Dominé and Shepson, 2002; Thomas et al., 2019), halogen compounds are also found within the snow. Sea-salt components, a source of halogens in snow in costal snowpack, might originate from migration from underlying sea-ice or from deposition of wind-transported sea-spray aerosol (Dominé et al., 2004). One characteristic of the cryosphere are its subfreezing temperatures and the consequent precipitation of chemical constituents at specific temperatures, their eutectic temperature, as also observed in sea-ice (Petrich and Eicken, 2009). It is known that the precipitations or phase changes of the reactants critically impact the reactivity (Bartels-Rausch et al., 2014; Kahan et al., 2014). Reduced chlorine production in frozen systems was observed by Wren et al. (2013) and by Sjostedt and Abbatt (2008), at temperatures below the eutectic temperature of sodium chloride in laboratory experiments that was attributed to the precipitation of sodium chloride. In an arctic field study, Custard et al. (2017) observed reduced $Cl_2$ production when temperatures dropped below the eutectic temperature of sodium chloride and suggested limited availability of chloride as consequence of the precipitation in this surface snow. Further, Lopez-Hilfiker and Thornton (2012) proposed the precipitation of sodium chloride to*

explain the changes in the production yield of ClNO2 from the reaction of N2O5 with saline frozen systems with temperature. More generally, it is not only the chlorine chemistry that responds to the phase of sodium chloride present in frozen systems both in the laboratory and in the environment. Oldridge and Abbatt (2011) showed that the rate of the heterogeneous reaction of ozone with bromide in sodium chloride -- water mixtures is strongly reduced once the salt precipitates below 252 K. The author explained this with the reduction in liquid volume that serves as reaction medium for the bromide in the sample due to the precipitation of sodium chloride."

Figure 5: Are optical images available for D-F as well? It would be particularly useful to refer to this, for example on Line 366 when the optical image is being described for D, for example. Also, please define the letters in the caption of Fig 5 so that reader is not required to refer back to Fig 3. In addition, consider changing the font on the letters and making them bold so that they are easier to discern; in particular, B is difficult to distinguish from B'.

*Unfortunately, no. This are all the pictures that we have. We will clearly state this in the caption in a revised manuscript and also apply the editing suggestions. Thank you.*

*"Figure 5: Optical microscopy pictures of the frozen samples. Each picture shows a 1 mm wide section of the sample holder with the sample. The letters refer to the samples in Figs. 3 and 4: A Solid NaCl at 248 K and 44 % relative humidity. B Ice with brine at 88 % RH and 259 K. Picture B' shows the deliquesced sample prior to freezing. C Sample below the eutectic temperature at 248-249 K in presence of ice. F Sample in absence of ice at a relative humidity lower than 73 % and at 244 K. Pictures of samples D and E were not taken."*

Additional Comments: - Fix section numbering throughout (all start with 1)
*Well, thanks. I must confess that I did not manage to get the numbering right. I have no idea what "Word" is doing here. I must leaf this to the editorial office to change in the final version.*

- Lines 37-38, 41, 44-45, 80-82: Please add references to these sentences.
*Done, thanks.*

«*Taken the abundance of chloride in the form of sea-salt over wide areas of the globe, the atmospheric chemistry of chlorine has long raised interest in a number of multiphase reactions that liberate chloride into chlorine species in the gas phase (Simpson et al., 2007; Finlayson-Pitts, 2010).*

*Common to all of these reactions is the generation of chlorine gases that either react with OH radicals or photolyze at wavelengths available in the troposphere to generate reactive chlorine (Fig. 1 and Finlayson-Pitts (2010)).*

*Its importance arises from its atmospheric abundance but also because its eutectic temperature of 252 K falls into typical springtime Arctic temperatures – a region and time period when atmospheric halogen chemistry is most active. For example, at the Arctic coast near Utqiaġvik (Alaska) temperatures between 247 K and 259 K have been reported (Custard et al., 2017). Another region of Earth's cryosphere, where temperatures drop below the 252 K is the troposphere (Thomas et al., 2019). Wang et al. (2015) has proposed a significant role of heterogeneous halogen chemistry on the ozone budget there. More recently, Murphy et al. (2019) has shown that the amount of sea-salt aerosol lifted to the upper troposphere is*

*small, casting some doubt on the environmental relevance of sea-salt as source of reactive halogens.»*

- Lines 57-65: This paragraph about bromine chemistry detracts from the focus of the current study and is suggested to be removed. Instead, it would be better to discuss the reaction mechanisms pertinent to chlorine chemistry shown in Figure 1.
*We will expand the discussion on the mechanism shown in Figure 1. This more detailed discussion of chlorine chemistry will then shift the focus away from bromine to chlorine. With all respect, we'd like to keep a somehow shortened mention of the bromine chemistry for two reasons. First, the findings in this work are not restricted to chlorine, but valid for any salt (at different temperatures compared to those found here for chlorine). Second, that chemistry at the interface differs from that in the bulk has, our judgement, been shown most elegantly for bromide. Therefore, we feel that this information is crucial to motivate investigation of the phase changes at an interface.*

- Figure 2: Please add the year to both Rumble citations in the caption and fix the spelling of "aqueous" in multiple locations the figure.
*Done, thanks.*

[Figure]

*«Figure 2: Phase diagram of the NaCl-water binary system. The data show the freezing point depression of sodium-chloride solutions (yellow filled circles) and give the concentration of an aqueous sodium chloride solution in equilibrium with ice in the temperature range of 273 K to 254 K (Rumble, 2019). The dark blue lines indicate the phase boundaries (Koop et al., 2000b; Rumble, 2019), that is it denotes the so-called liquidus and solidus line, respectively, and thus*

*shows the temperature and concentration range where ice and aqueous sodium chloride solution co-exist. The eutectic temperature of sodium chloride – water binaries is 251.9 K (Koop et al., 2000a). Also shown is a typical experimental procedure (red arrows and cross).»*

- Line 228: Fix typo "disused".
*Done, thanks.*

- Figure 4: Consider making the phase labeling on plot A more similar to Fig 2. For example, the labeling of the "ice melting" line was confusing at first given the locations of the individual words surrounding the line in the figure.
*Done, thanks. We have updated Figure 2, where the phases are now no longer labelled near the lines.*

- Lines 348-359: Consider moving this paragraph to the methods section, as it de- scribes how the experiment was conducted, rather than the results of the experiment.
*Yes, we will better separate the experimental description from discussing the results. Based on the suggestions by referee 3 we will slightly re-structure the manuscript to better link results in Figure 3 to the conditions of the individual measurements and also omit repetitions of the experimental procedures.*

*We have expanded this section by adding more observables*

*«In a typical experiment, anhydrous salt was exposed to increased relative humidity at a fixed temperature of 259 K. Once the relative humidity reached 72 %, the sample started to dissolve by water up-take from the gas-phase and an aqueous solution was formed (brine). This phase change was evident by the sample becoming shiny and then forming transparent spheres as observed by an endoscope digital camera (Fig. 5B'). Then, the relative humidity was further increased to cross the ice stability line until ice nucleation occurred at a modest supersaturation of typically 90 % to 95 % relative humidity at 259 K. Ice nucleation was evident by a sharp pressure drop from the pressure dosed to the cell to the water vapor pressure of ice at that temperature, for example 88 % relative humidity at 259 K. In some experiments, temperature was lowered 1 K to 2 K as well to trigger ice nucleation. Please note, that temperature was always well above the homogenous freezing temperature, which was found at 210 K to 230 K at relative humidities of 60 % to 90 % (Koop et al., 2000a).»*

*and by moving the more general description to the experimental part:*

*«Water vapour was dosed to the flow-through cell via a 0.8 mm i.d. steel capillary from the vapour above liquid water (Fluka Trace Select 142100-12-F) in a vacuum-sealed, temperature-controlled glass reservoir. Before dosing, the water was degassed by 4 freeze-pump-thaw cycles. The the water dosing and thus the partial pressure or relative humidity that the sample is exposed to  was varied by setting the temperature of the reservoir to change the flux of water vapour into the experimental cell.»*

- Line 370: Please provide the temperature here in parentheses for clarity (rather than just simply 11 K below the eutectic) to aid the reader in referring to Fig 4 and quickly finding the proper star marker.
*Done, thanks.*

- Line 371: Where is this "11.4 K below the eutectic" data shown? This sentence seems like it is discussing the current work, but Fig 3 only shows 10 K and 12 K below the eutectic.

*«Consistent with that, the chloride has a local environment indistinguishable from that of the hydrohalite 11.4 K below the eutectic temperature and in the presence of ice in the current study (Fig. 3D)».*

*Corrected, thanks. The temperature given in Figure 3 were wrong. Apologies. We have also added the partial pressure of water for each NEXAFS spectrum to the caption.*

[Figure]

Figure 3: Partial electron yield chlorine K-edge NEXAFS spectra of the sodium chloride -- water binary system: A Solid NaCl at 248 K and 44 % relative humidity (0.34 mbar water vapour pressure). B Aqueous NaCl solution in equilibrium with ice at 88 % RH (1.82 mbar) and 259 K. C An averaged spectrum at the thermodynamic ice stability line at 248 K to 249 K and 78 % (0.60 mbar) to 80 % RH (0.71 mbar). D An individual spectrum upon further cooling to 241 K and 74 % RH 0.29 mbar). E An individual spectrum upon heating back to 249 K in the ice stability domain at 79 % RH (0.69 mbar). F The averaged spectrum at 244 K and a RH of 59 % (0.32 mbar) and 73 % (0.40 mbar), lower than the ice stability domain. See Fig. 4 for precise measurement settings. The shaded area in the colour of the graph in C and F denote the standard deviation of 3 and 2 repeated NEXAFS acquisitions. Also shown are NEXAFS spectra of NaCl salt and aqueous solutions for comparison that were detected in fluorescence mode and not in partial electron yield (Huggins and Huffman, 1995). The brownish shaded area (I-III) highlights regions in the NEXAFS spectra discussed in the text. The grey shaded area at 2821 eV highlights the photon energy region where carbon-chlorine bonds from carbon contamination might show an absorption feature (see text for details).

- Lines 378-383: Is it possible that the hydrohalite may form within the bulk prior to the surface, explaining the higher temperature observed by Malley et al (2018) compared to this work?
*Good point, thanks. We will add this to the manuscript.*

*"It appears thus that the precise occurrence of crystallisation is governed by stochastics at the surface as has been shown for freezing of bulk samples (Alpert and Knopf, 2016). Because of the good agreement between the precipitation temperatures observed in this study and in (Koop et al., 2000a), we believe that the deviation from (Malley et al., 2018)'s results does not indicate differences in the freezing behaviour at the surface vs. in the bulk."*

- Line 387: For clarity, I suggest adding "at 5 K below the eutectic temperature" after "spectrum" in this sentence.
*Done, thanks.*

- Line 392: Change "snow" to "ice" here, since authentic snow was not studied in this work.
*Done, thanks.*

- Lines 394-426: This is a very long paragraph. Please consider breaking up.
*Done, thanks. We also added some guiding sentences for the reader.*

*For samples that were cooled to temperatures that triggered efflorescence, the chlorine NEXAFS spectra show that the hydrohalite is the dominating phase at the interface of frozen sodium chloride – water binary mixtures.  Cho et al. (2002) have shown that when frozen aqueous solutions were warmed, a liquid fraction was observed below the eutectic temperatures.  In their experiments, ice was frozen in NMR tubes lowering the temperature to 228 K in 15 min. which is significantly colder than the efflorescence temperatures observed here and by Koop et al. (2000a). After 10 minutes, the samples were warmed and NMR signals were recorded. Interestingly, Cho et al. (2002) have observed the liquid fraction only in experiments where the sodium chloride concentration in the initial aqueous solution was below 0.01 mol l-1. If the initial aqueous solution had a concentration of 0.5 mol l-1 no indication of liquid features below the eutectic were found. Tasaki et al. (2010) has shown a similar concentration dependence for sodium bromide solutions using X-ray absorption reporting solvated bromide in the bulk of the samples below the eutectic temperature only for concentrations below 50 mmol l-1.*

*We will now detail the concentration of brine in the study presented here to elaborate if differences in concentration might explain the differences the observed liquid content of sub-eutectic samples. The experiments described here started with an aqueous solution that was formed in-situ and was kept in equilibrium with a vapour pressure of roughly 1.9 mbar. The chloride concentration in such solutions is close to the concentration in a solution at 1.8 mbar and at 259 K, where ice nucleation occurred and where the freezing point depression data*

*give a concentration of 3.5 mol l-1. This concentration can be directly compared to the concentration in the initial solutions of Cho et al. (2002), which ranged from below 0.01 mol l-1 to 0.5 mol l-1. This back-of-the-envelope calculation thus suggests that the concentration of the solutions from which ice nucleated in the experiments reported here exceeded those described by Cho et al. (2002) for which no liquid fraction was observed.*

*Next, we discuss how higher concentrations of initial solution might impact the location of brine in the frozen matrix. The concentration of the initial solution from which ice precipitated, determines the ice to brine ratio after ice formation. This is because the volume of the brine relative to that of ice is given by the water to sodium chloride ratio in the initial solution. The concentration of the brine is a sole function of temperature, and not of the initial concentration of the solution.*

- Lines 428 – 439: This paragraph overall should be revised to make it clearer and easier to read and relate the previous work to the current results. In particular, the goal of this paragraph could be clarified at the beginning of the paragraph to help guide the reader, as I had to read the beginning sentences multiple times to understand them in the context of the current work.
*Done, thanks.*

*The previous argumentation is based on the features in the NEXAFS spectrum of sodium chloride – ice mixtures shown in Fig. 3 D and E being dominated by the NEXAFS spectrum of hydrohalite shown in Fig. 3F. In particular the spectrum in Fig. 3E, acquired 3 K below the eutectic temperature, shows a shoulder starting at 2823 eV.  Such a feature is absent in the spectrum of the hydrohalite (Fig. 3 F), but the spectrum of brine (Fig. 3 B) shows an increase in absorption starting at this X-ray energy. We can thus not exclude the presence of brine in the samples where the hydrohalite dominates the NEXAFS.  Taken the spectra quality and the small difference in the shape of the liquid and of the hydrohalite spectrum, it is beyond the scope of this work to elaborate whether the NEXAFS spectrum in Fig. 3E might be understood by deconvoluting it in its hydrohalite and brine components and by this reveal a fraction of the chloride being embedded in a brine-like hydrogen bonding network. Two reasons might explain the presence of liquid in these samples at sub-eutectic temperatures. First, one might expect a certain distribution in the size of micropockets and a small fraction of the pockets might thus be small enough to stabilize liquid at these temperatures. This explanation is consistent with the sample at warmer temperatures showing a more intense shoulder at 2823 eV. Secondly, some of the chloride might form solvation shells with water molecules from the ice matrix as proposed for trace gases adsorbed to ice {Krepelova; Bartels-Rausch, 2019}. In particular, we have recently reported chloride forming solvation shells in the interfacial region of ice upon adsorption of HCl at 253 K (Kong et al., 2017). The surface concentration as derived from XPS suggested that it was done in the stability domain of ice, i.e., the concentration of HCl was too low to melt the ice. Oxygen K-edge NEXAFS spectra showed that a substantial fraction of the water molecules at the air-ice interface is arranged in a hydrogen-bonding structure like that of liquid water.We like to note, that Krepelova et al. (2010a)'s oxygen K-edge spectra of sodium chloride -- ice did not reveal water molecules being coordinated like in the liquid. Taken together the signal from the hydrohalite by far exceeds the signal from a chloride in brine or in an liquid-like environment at the molecular level.*

- Lines 444-445: This sentence appears to be missing its end.
*We have rewritten this paragraph.*

*«Such a trajectory, that is the temperature and water vapor pressure the sample experienced, is shown in Fig. 4C (green solid line). In this set of experiments, water was evaporated by decreasing the relative humidity to about 70 % at 252 K from a brine sample in absence of ice, followed by lowering the temperature to 247 K at constant partial pressure of water (so that the relative humidity increased to about 80 %).»*

- Line 450: I believe the authors mean to refer to Fig 3A here.
*Corrected, thanks.*

- Line 475: Please provide an estimate or approximate range here in parentheses to provide improved understanding of what "the upper few nanometre" mean.
*Done, thanks.*

- Line 447: I suggest changing "identical" to "similar" here, as the sea salt aerosol in the environment are more complex than simple NaCl-H2O systems. Of particular relevance is that sea spray aerosol particles can have thick organic coatings (e.g., Kirpes et al. 2019, ACS Central Science).
*Agree. We will change this and mention organic coatings, thanks.*

*Introduction: "An advantage of this experimental approach with environmental relevance is that the relative humidity precisely matches that in the atmosphere in contact and in thermodynamic equilibrium with ice clouds or snow cover, because the relative humidity is a sole function of temperature in presence of ice. Therefore, the chemical concentration of such particles exactly matches those of same composition in snow or in the atmosphere under environmental conditions."*

*Line 447: « The sample was always in equilibrium with gas-phase water which makes cooling conditions identical and concentration of brine similar to that of aerosol particles embedded in snow or in the troposphere.»*

*«From temperature and relative humidity alone, nucleation or efflorescence cannot be predicted at the interface as in the bulk. We suggest that further studies focus on samples with more complex chemical composition to enhance our knowledge of environmental multiphase chemistry. For example, organic compounds are a common constituent of sea-salt aerosol (O'Dowd et al., 2004) { Kirpes, 2019} and recently we have shown how there presence impacts the microphysics and thus reactivity of salt particles towards ozone (Edebeli et al., 2019).»*

- Lines 489 – 490: It would be useful to add discussion about the temperature ranges that are important to consider here (that would matter) when considering the polar environment that is being discussed in which temperature swings regularly occur with changing weather.

A more detailed discussion referring to temperature ranges would be helpful to bring the gap to observations, based on the temperature and RH- dependent results of the current work.

*Good point, thanks. We will extend the paragraph in the introduction on typical temperature ranges in Polar environments:*

*"Its importance arises from its atmospheric abundance but also because its eutectic temperature of 252 K falls into typical springtime Arctic temperatures – a region and time period when atmospheric halogen chemistry is most active. For example, at the Arctic coast near Utqiaġvik (Alaska) temperatures between 247 K and 259 K have been reported (Custard et al., 2017). Another region of Earth's cryosphere, where temperatures drop below the 252 K is the troposphere (Thomas et al., 2019). Wang et al. (2015) has proposed a significant role of heterogeneous halogen chemistry on the ozone budget there. More recently, Murphy et al. (2019) has shown that the amount of sea-salt aerosol lifted to the upper troposphere is small, casting some doubt on the environmental relevance of sea-salt as source of reactive halogens. "*

*And come back to this in the Summary*

*"Multiphase reactions may proceed at accelerated rates in these highly concentrated brines at temperatures ~10 degrees below the eutectic compared to reactions on solid hydrohalite. This temperature range of ~240 K is frequently observed in polar coastal sites during spring and also in the free troposphere."*

- Lines 495-496: This discussion of micro-pockets is confusing when comparing to text on Line 424. Please clarify.
*We will reword to make this clearer :*

*«We suggest that the brine observed by Cho et al. (2002) is a consequence of the presence of very small pockets holding the brine. Because pockets only tend to be small enough to establish a significant depression in freezing point when solutions with low concentration are freezing, and because aerosol at typical relative humidity that prevail in cold parts of the atmosphere are highly concentrated; we suggest that such micro-pockets at the air-ice interface are of small relevance to the environment.»*

- Lines 500-501: Would the concentration effect discussed on page 17 have an impact here?
*Yes, this might well be; depending on the source and location of the sodium chloride in sea ice (from ocean water with lower salinity, or from deposited aerosol in snow on sea-ice). We prefer not to go into detail here.*

- Lines 503-504: Perhaps this would also have an impact on brine migration upward through the snow (i.e. Domine et al. 2004, Atmos. Chem. Phys.)?
*Interesting aspect, thanks.*

*«For modern Earth, precipitation of brine constituents in sea ice is relevant for ion mobility and might result in ion fractionation during wash out events (Maus et al., 2008; Obbard et al., 2009; Maus et al., 2011) and possibly brine migration upwards through the snow {Domine, 2004}.»*

- Lines 506-510: It would be useful to add discussion here about where in the atmo- sphere this might matter (using the temperature and RH knowledge from this work). Also, would the history of the production of sea spray aerosol as droplets (and therefore not starting at 0% RH) matter in terms of halite vs hydrohalite based on the results presented herein? Also, might the presence of organics coating the sea salt aerosol have an impact?

*Thank you for pointing to the atmospheric relevance. It does certainly matter in arctic spring, when snowpack chemistry is most active. We will add a paragraph in the discussion to elaborate this and other regions in more detail.*

*«Precipitation of sodium chloride in sea spray aerosol in the troposphere, not embedded in snow or sea ice, is further of ongoing interest. Again, the arctic coastal areas are of high relevance here, because temperature and relative humidity frequently favour precipitation of hydrohalite. Other studies have focus on the temperature range of 230 K to 260 K at a relative humidity of 30 % to 70 % (see Fig. 4 and references therein) to explore the precipitation in the dryer free troposphere.  focus is placed on whether anhydrous sodium chloride (halite) or hydrohalite precipitates.  In regions of the phase diagram, where the hydrohalite is the thermodynamic stable form, precipitation of the halite was observed with impacts on stability of the solid phase upon warming and or humidification, since the deliquesce relative humidity of the two compounds differs by 6 percentage points (Wagner et al., 2012; Wise et al., 2012; Peckhaus et al., 2016).»*

*The history of the droplets certainly matters. We believe that this is a strength of the current study. Indeed, the sample was exposed to UHV (0 % RH) prior to adding gas-phase water. The important point is that at the time we started to cool and approach the nucleation temperature the salt had formed a brine solution in equilibrium with the water vapor at RH identical to that above ice. The nucleation experiments thus started with samples that were liquid and identical in concentration to those found in/on snow.*

*Organic coatings might indeed impact the surface tension and thus the freezing properties in nano pockets or even act as antifreeze proteins. This is an interesting point, worth further studies which we will happily suggest in a revised version.*

---

## Author Comment (AC3) · 19 Feb 2021

Bartels-Rausch et al. present an original piece of research, investigating the phase changes in the NaCl-water binary system by measuring NEXAFS spectra across the chlorine K-edge when varying the temperature- and relative humidity conditions the particles are exposed to. The assignment of the three basic spectra types recorded for anhydrous NaCl particles, aqueous NaCl solution droplets, and the first-time mea- surement for hydrohalite, are convincing. I do not have any major concerns regarding the scientific content and quality of the manuscript, but I would strongly encourage the authors to improve to some extent the way of presentation of the data, and thereby better "guide the reader" through the various parts of the article.

*We thank the referee for the supporting judgement of the manuscript.*

Major comment:

All the while reading the section "NEXAFS of brine, halite, and hydrohalite", I kept won- dering how exactly the experiment was carried out, i.e., what was the actual trajectory in the T-RH space, how exactly was the freezing of the NaCl solution droplets induced, how did you move along the ice/NaCl(aq) equilibrium line, how was hydrohalite with- out ice (Fig. 3F) formed? All these experimental details are only provided in later parts of the article (e.g. line 348ff, line 442ff). I see some good reason for the chosen manuscript structure, i.e., that you first want to describe the NEXAFS spectra, discuss some technical aspects like spectra quality and reproducibility, compare your measure- ments with literature spectra - and then later discuss the exact formation conditions for hydrohalite and the atmospheric implications based on the T-RH trajectory. But having a better general idea of the experimental procedure before reading the section with the NEXAFS spectra would be in my opinion a clear improvement regarding the clarity of presentation. One suggestion would be the following: On line 84, you introduce the NaCl-water phase diagram, but describe it with some "hypothetical" trajectory, starting from a sample below 251.9 K and then increasing the temperature. But why not discuss the phase diagram with a "proper" trajectory from your experiments, which could be schematically depicted in Fig. 2 – meaning you start with an aqueous NaCl solution droplet, induce some supercooling to nucleate ice, and then move along the liquidus curve towards the eutectic and below, to investigate at which point hydrohalite precipitates. This is also the trajectory during which most of the spectra shown in Fig. 3 (B – E) were recorded (apart from the anhydrous NaCl, A, and the "ice-free" hydrohalite, F. At the end of the introduction or at the beginning of the "NEXAFS of brine, halite, and hydrohalite" section, you should then include a paragraph and inform the reader about the general structure of the manuscript, i.e., that you want to disentangle the detailed description of the NEXAFS spectra of the three species (shown first) from the detailed analysis of the phase changes when moving in the T-RH space (shown later). Please also number all section and subsection headings correctly.

*This is a splendid idea that we will happily follow. Thank you. Figure 1 and the paragraph describing it will be updated as follows:*

[Figure]

*«Figure 2: Phase diagram of the NaCl-water binary system. The data show the freezing point depression of sodium-chloride solutions (yellow filled circles) and give the concentration of an aqueous sodium chloride solution in equilibrium with ice in the temperature range of 273 K to 254 K (Rumble). The dark blue lines indicate the phase boundaries (Koop et al., 2000b; Rumble), that is it denotes the so-called liquidus and solidus line, respectively, and thus shows the temperature and concentration range where ice and aqueous sodium chloride solution co-exist. The eutectic temperature of sodium chloride – water binaries is 251.9 K (Koop et al., 2000a). Also shown is a typical experimental procedure (red arrows and cross).»*

*"While the phase diagram of sodium chloride – water binary mixtures and the thermodynamic stability domains of salt, solution, and ice are well known (Koop et al., 2000a), the precise occurrence of nucleation and sodium chloride precipitation is still debated (Koop et al., 2000a; Wise et al., 2012; Peckhaus et al., 2016). Figure 2 shows a part of phase diagram of sodium chloride - water mixtures and can be used to illustrate the appearance of the individual phases. Below 251.9 K, the eutectic temperature of sodium chloride (Koop et al., 2000a), solid sodium chloride dihydrate (hydrohalite, NaCl•2H2O) and solid water (ice) are the energetically favoured phases. Between the eutectic temperature and the so-called liquidus line, ice and sodium chloride solution (brine) co-exist. The ice will melt completely above the liquidus line, and an aqueous sodium chloride solution is the only phase present. The focus of this work was to experimentally observe phase changes of sodium chloride below the eutectic temperature. A typical experimental procedure started with a dry sample of anhydrous sodium chloride (halite, NaCl) which was exposed to increasing gas-phase water at constant temperature of 259 K. By absorbing water from the surrounding air, a phase transition from the solid salt to a liquid solution (deliquescence) took place. Upon increasing the gas-phase water dosing further (Fig. 2, red arrow) ice crystalised and a two-phase system of ice and brine occurred (Fig. 2, red cross). After probing the sample at this*

*position in the phase diagram (see below), temperature was lowered and the dosing of the water-vapor adopted to move along the liquidus line to below the eutectic temperature to perform additional measurements. During this cooling period, salt concentration ̲a̲n̲d̲ ̲v̲o̲l̲u̲m̲e̲ ̲o̲f̲ ̲t̲h̲e̲ ̲b̲r̲i̲n̲e̲ ̲c̲h̲a̲n̲g̲e̲s̲.̲ ̲S̲u̲c̲h̲ ̲c̲h̲a̲n̲g̲e̲s̲ ̲w̲i̲t̲h̲ ̲v̲a̲r̲y̲i̲n̲g̲ ̲r̲e̲l̲a̲t̲i̲v̲e̲ ̲h̲u̲m̲i̲d̲i̲t̲y̲ ̲(̲h̲y̲g̲r̲o̲s̲c̲o̲p̲i̲c̲ ̲g̲r̲o̲w̲t̲h̲)̲ ̲h̲a̲v̲e̲ ̲l̲o̲n̲g̲ ̲b̲e̲e̲n̲ ̲d̲i̲s̲c̲u̲s̲s̲e̲d̲ ̲f̲o̲r̲ ̲a̲e̲r̲o̲s̲o̲l̲ ̲i̲n̲ ̲t̲h̲e̲ ̲t̲r̲o̲p̲o̲s̲p̲h̲e̲r̲e̲.̲ In other words,…"*

Additional comments:

1) Regarding line 88 and 112: Isn't 251.9 K the ice-hydrohalite eutectic, and shouldn't ice and NaCl*2H2O be the energetically favored phases below that temperature?

*Yes, you are correct. Apologies for the confusion. In this early part of the manuscript, we tried to mention "sodium chloride" in the sense of a salt of sodium chloride (halite or hydrohalite). We will be careful to me more specific in the revised version.*

*Line 88: «Oldridge and Abbatt (2011) showed that the rate of the heterogeneous reaction of ozone with bromide in sodium chloride -- water mixtures is strongly reduced once the salt precipitates below 252 K.»*

*Line 112: «Below 251.9 K, the eutectic temperature of sodium chloride (Koop et al., 2000a), solid sodium chloride dihydrate (hydrohalite, NaCl•2H2O) and solid water (ice) are the energetically favoured phases.»*

2) Line 279/280: "249 K in the presence of ice and at 244 K in the absence of ice": When reading this sentence for the first time, I was also wondering whether spectra 3E and 3F were from the same trajectory and asked myself how the experimental procedure could have been – it is only explained much later (line 441ff). You should include here at least a short description of how the particles from spectrum F were generated.

*Thanks, done. We have added a sentence stating the relevance of spectra in Figure 3C-E and separated the discussion of the spectra in Fig. E and F:*

*Line 245: "Figure 3C-E show NEXAFS spectra acquired in the ice stability domain at temperatures below the eutectic temperature of 251.9 K (Koop et al., 2000a). By comparing these spectra to the spectra in Fig. 3 A-B, the phase of the sodium chloride in presence of ice below the eutectic temperature will be discussed."*

*Line 279: "Figure 3E shows a spectrum after warming the sample back to 249 K in the presence of ice. Clearly, the observed shape in region I show that hydrohalite and not liquid brine is the dominant phase of this sample. In Fig. 3 F a spectrum at 244 K in absence of ice is shown, again the shape is in good agreement to spectrum shown in Figure 3D."*

3) Line 337: Here starts the detailed discussion of the phase behavior of the particles in the T-RH space. I would also appreciate an introductory paragraph describing how this section is structured and what different aspects are discussed. Otherwise, the reader may quickly lose track of things. For example, the heading "Liquid below eutectic and nucleation" in line 347 comprises a very long section that could be divided into various subsections. In the headings, you could also be more specific what you mean by "nucleation", nucleation of ice or the precipitation of hydrohalite.

*Thanks, done.*

*«Now that we have identified halite, the hydrohalites, and the aqueous solution by means of the NEXAFS spectra at the interfacial region, we discuss their observation in the phase diagram. Generally, we have observed solid sodium chloride as halite or hydrohalite at temperatures below 240 K and at 44 % to 79 % RH and brine in the temperature range of 248 K to 259 K and RH of 78 % - 88 %.*
*Interestingly, the NEXAF spectra have revealed the dominant presence of brine in one sample and of hydroahliate in other samples. All these samples were probed below the eutectic temperature indicating that not only the temperature and relative humidity, but also the trajectory to reach these settings (or the history of the sample) might determine the phase. Therefore, we will detail the humidity and temperature history of each sample in the following in detail. Also, we will compare our findings to the extensive literature work of observed phases in presence and absence of ice. This discussion will be based on the sodium chloride – water phase diagram in the temperature – relative humidity space (Figure 4A) as initially constructed by Koop et al. (2000a).»*

4) Line 353: Can you please quantify "modest" supersaturation – did you need to reach the homogeneous freezing limit for aqueous solution droplets (Koop et al., 2000b) or did the surface catalyze heterogeneous ice nucleation?

*Thanks, for mentioning homogeneous freezing. We will certainly add this detail to a revised version. The homogeneous freezing temperatures reported by Koop at 60-90% RH are about 210-230K, lower than the temperature range of this study.*

*"Then, the relative humidity was further increased to cross the ice stability line until ice nucleation occurred at a modest oversaturation of typically 90 % to 95 % relative humidity at 259 K. Ice nucleation was evident by a sharp pressure drop from the pressure dosed to the cell to the water vapor pressure of ice at that temperature, for example 88 % relative humidity at 259 K. In some experiments, temperature was lowered 1 K to 2 K as well to trigger ice nucleation. Please note, that temperature was always well above the homogenous freezing temperature, which was found at 210 K to 230 K at relative humidities of 60 % to 90 % {Koop, 2002}. "*

5) Line 442, regarding the trajectory when recording spectrum 3F: Could you please elaborate a bit more on the idea and temporal order behind this trajectory (Fig. 4C), I did not quite understand how the procedure was – did you again try to cool a NaCl droplet along the liquidus curve but without inducing sufficient supersaturation to nucle- ate ice? And then at about 244 K reduced the RH to 59% to induce the crystallization of hydrohalite? And then increased RH back to 73%

*Apologies for being not clear. The reasoning behind this experiment was to observe hydrohalite in absence of ice. In this set, we started with brine in absence of ice by evaporating the water (decreasing RH to ~70%) at 252K. then we lowered the temperature to 247K at constant partial pressure of water (so the RH increases to > 80%). We crossed the ice stability line, but supersaturation was not sufficient to nucleate ice. We repeated this procedure to reach the RH-T positions "F".*
*We will elaborate this procedure in more detail in the revised manuscript.*

*Hydrohalite can also precipitate in absence of ice by evaporating water from a solution at temperatures below 273 K (Craig et al., 1975; Yang et al., 2017). Such a trajectory, that is the temperature and water vapor pressure the sample experienced, is shown in Fig. 4C (green solid line). In this set of experiments, from a brine sample in absence of ice at 252 K (above the eutectic temperature) water was evaporated by decreasing the relative humidity to about 70 % at 252 K, followed by lowering the temperature to 247 K at constant partial pressure of water (so that the relative humidty increased to about 80 %). When the ice stability line in the phase diagram was crossed, ice nucleation was not observed as the oversaturation was not sufficient to trigger ice nucleation. This procedure was repeated to further lower the temperature to 244 K at 59 % relative humidity. Then, the first NEXAFS was recoreded at 244 K and 59 % RH (Fig. 4, green diamond), where in absence of ice nucleation was visually observed. The location in the phase diagram is in agreement with Wagner et al. (2012)'s observation of salt deposits in aerosol droplets in an aerosol chamber (AIDA)in absence of ice. The second NEXAFS spectra resembling that of the hydrohalite and in absence of ice was recorded at a slightly higher relative humidity of 72 % (Fig. 4, green diamond). Both NEXAFS spectra (Fig. 4, green diamonds) were identified as hydrohalite (Fig. 3F). The sample at 44 % RH (Fig. 4C, blue line) has been exposed to 0 % RH prior to acquiring the NEXAFS spectrum (Fig. 4, blue square), which removes the crystal water (Light et al., 2009; Wise et al., 2012).*

technical corrections:

*Thanks, all technical correction are done.*

1) Doesn't the title sound a bit awkward? Maybe better: "Investigation of interfacial supercooling . . ." or ". . . NaCl solutions studied by X-ray absorption spectroscopy"
2) Line 86: "shows a part of the phase diagram"
3) Line 109: "in equilibrium with ice"
4) Line 115 – 117: Very long sentence, please split into two.
5) Line 138: "nitrate and chloride form solvation cells"
6) Line 173: "and with a pass energy"
7) Line 186: "take-off angle of detected electrons (?)"
8) Line 220: photon energy in eV
9) Line 240: Br- (superscript is missing)
10) Line 338: "the hydrohalite"
11) Line 353: maybe better "supersaturation"
12) Line 355: maybe: "and is thus a sole function"
13) Line 411/412: This is a pretty nested sentence, please re-phrase.
14) Line 416: "in larger patches/inclusions"
15) Line 444/445: The sentence seems incomplete, it also misses the point at the end.
16) Line 450: Fig. 3A
17) Line 462: check super- and subscripts

---

## Editor Comment (EC1) · Florent Dominé (Editor) · 23 Feb 2021

Dear Authors,

Thank you for your constructive responses to the reviewers' comments. I encourage you to submit a revised version along the lines detailed in your replies.

Best regards

Florent Domine
* * *